# Structural basis underlying the synergism of NADase and SLO during group A *Streptococcus* infection

Wei-Jiun Tsai[1], Yi-Hsin Lai[1,18], Yong-An Shi[2,18], Michal Hammel [3], Anthony P. Duff[4], Andrew E. Whitten [4], Karyn L. Wilde [4], Chun-Ming Wu[5], Robert Knott[4], U-Ser Jeng[5,6], Chia-Yu Kang[7], Chih-Yu Hsu[8], Jian-Li Wu[9], Pei-Jane Tsai[1,8,10], Chuan Chiang-Ni [2,11,12,13], Jiunn-Jong Wu[14,15], Yee-Shin Lin[7,10], Ching-Chuan Liu[7], Toshiya Senda [16] & Shuying Wang [1,7,10,17✉]

Group A *Streptococcus* (GAS) is a strict human pathogen possessing a unique pathogenic trait that utilizes the cooperative activity of $NAD^+$-glycohydrolase (NADase) and Streptolysin O (SLO) to enhance its virulence. How NADase interacts with SLO to synergistically promote GAS cytotoxicity and intracellular survival is a long-standing question. Here, the structure and dynamic nature of the NADase/SLO complex are elucidated by X-ray crystallography and small-angle scattering, illustrating atomic details of the complex interface and functionally relevant conformations. Structure-guided studies reveal a salt-bridge interaction between NADase and SLO is important to cytotoxicity and resistance to phagocytic killing during GAS infection. Furthermore, the biological significance of the NADase/SLO complex in GAS virulence is demonstrated in a murine infection model. Overall, this work delivers the structure-functional relationship of the NADase/SLO complex and pinpoints the key interacting residues that are central to the coordinated actions of NADase and SLO in the pathogenesis of GAS infection.

[1] Institute of Basic Medical Sciences, College of Medicine, National Cheng Kung University, Tainan, Taiwan. [2] Graduate Institute of Biomedical Sciences, College of Medicine, Chang Gung University, Taoyuan, Taiwan. [3] Molecular Biophysics and Integrated Bioimaging, Lawrence Berkeley National Laboratory, Berkeley, CA, USA. [4] Australian Nuclear Science and Technology Organisation, Lucas Heights, NSW, Australia. [5] National Synchrotron Radiation Research Center, Hsinchu Science Park, Hsinchu, Taiwan. [6] Department of Chemical Engineering, National Tsing Hua University, Hsinchu, Taiwan. [7] Department of Microbiology and Immunology, College of Medicine, National Cheng Kung University, Tainan, Taiwan. [8] Department of Medical Laboratory Science and Biotechnology, National Cheng Kung University, Medical College, Tainan, Taiwan. [9] Institute of Biological Chemistry, Academia Sinica, Taipei, Taiwan. [10] Center of Infectious Disease and Signaling Research, National Cheng Kung University, Tainan, Taiwan. [11] Department of Microbiology and Immunology, College of Medicine, Chang Gung University, Taoyuan, Taiwan. [12] Molecular Infectious Disease Research Center, Chang Gung Memorial Hospital, Linkou, Taiwan. [13] Department of Orthopedic Surgery, Chang Gung Memorial Hospital, Linkou, Taiwan. [14] Department of Medical Laboratory Science and Biotechnology, Asia University, Taichung, Taiwan. [15] Department of Medical Research, China Medical University Hospital, China Medical University, Taichung, Taiwan. [16] Structural Biology Research Center, Institute of Materials Structure Science, High Energy Accelerator Research Organization (KEK), Tsukuba, Ibaraki, Japan. [17] Department of Biotechnology and Bioindustry Sciences, College of Bioscience and Biotechnology, National Cheng Kung University, Tainan, Taiwan. [18] These authors contributed equally: Yi-Hsin Lai, Yong-An Shi. ✉email: sswang23@mail.ncku.edu.tw

Group A *Streptococcus* (GAS) is a human-exclusive pathogen renowned for its highly aggressive destruction of host tissues and responsible for more than 500,000 deaths annually[1]. GAS infections can lead to diverse clinical manifestations, ranging from mild impetigo and pharyngitis to life-threatening scarlet fever, streptococcal toxic shock syndrome, and necrotizing fasciitis[1,2]. An important pathogenic feature of GAS is to trigger cytotoxicity in host cells that can facilitate bacteria to invade deeper tissues[3]. It is suggested that the capability of GAS to remain a successful pathogen is correlated with its considerable number of secreted products[4]. Two secreted toxins, NAD$^+$-glycohydrolase (NADase) and Streptolysin O (SLO), play central parts in invasive GAS infections[5–8] and correspond to the emergence of the globally disseminated GAS M1 lineages[9–12]. The presence of both toxins can synergistically enhance GAS cytotoxicity and intracellular survival[13–16]. Moreover, it is suggested that direct interaction and mutual stabilization between NADase and SLO are crucial for enhancing GAS virulence[17], signifying the importance of the NADase/SLO complex formation in GAS pathogenesis.

NADase and SLO are functionally-interdependent during GAS infection. SLO is a pore-forming toxin belonging to the family of cholesterol-dependent cytolysins (CDCs) and can disrupt eukaryotic cell membranes with its cytolytic activity[18,19]. Disruption of the host membrane by SLO pores leads to the deaths of macrophages, neutrophils, epithelial cells, and red blood cells[14,20–23]. In addition to acting as a typical member of CDCs, SLO performs a distinct membrane binding mode through association and cooperation with its cotoxin NADase to promote pore-formation during GAS infection[24,25]. SLO is required for the delivery of the effector NADase into the host cytosol in a pore-formation-independent process termed cytolysin-mediated translocation (CMT)[26,27]. The translocated NADase exerts its cytotoxic effects by hydrolyzing β-NAD$^+$ into nicotinamide and ADP-ribose, resulting in cell death through β-NAD$^+$ depletion, and promotion of the intracellular survival of GAS within macrophages, epithelial cells, and endothelial cells[14,15,28–31]. Moreover, the cooperative activity of NADase and SLO enables GAS to escape immune responses and survive within host cells by inhibiting the maturation of phagolysosome and xenophagic machinery[14,15,21,23].

The enzymatic activity of NADase is important in GAS virulence. However, many GAS variants deficient in NADase activity are found among clinical isolates, and these variants exhibit comparable cytotoxicity to the NADase-proficient strains[32–34]. A common polymorphism G330D, with the presence of aspartate instead of glycine at position 330 of NADase, lacks detectable NADase activity[34]. However, NADase$_{G330D}$ remains a potent virulence factor and is able to interact with SLO similarly to the wild type[17], suggesting the molecular evolution of NADase has a tendency for preservation of the NADase/SLO complex.

The synergistic activities of NADase and SLO through physical association is a unique example among Gram-positive bacteria, as secreted toxins generally function exclusively. Substantial progress has been made to understand the coordinated activities and functional interdependence between NADase and SLO. However, the manner in which NADase interacts with SLO to exert the synergistic effects is a long-standing question. Here, the structure of the NADase/SLO complex is presented, obtained using X-ray crystallography and small-angle scattering. The atomic details allow us to identify key interacting residues and demonstrate the biological importance of the NADase/SLO complex to the pathogenesis of GAS infection in vitro and in a mouse model.

## Results

**Crystal structure of the NADase$_{193-451,G330D}$/SLO$_{106-574}$ complex.** To reveal atomic interactions between NADase and SLO by

**Table 1 Crystallographic data and refinement statistics.**

| NADase$_{193-451,G330D}$/SLO$_{106-574}$ complex (PDB 7WVH) | |
|---|---|
| Data Collection | |
| Space group | C2 |
| Cell dimensions | |
| a, b, c, (Å) | 246.09, 52.48, 169.40 |
| α, β, γ (°) | 90.0, 132.04, 90.0 |
| $R_{merge}$ | 0.04 (0.54)$^a$ |
| $I/\sigma I$ | 31.6 (2.4) |
| Completeness (%) | 99.8 (100.0) |
| Redundancy | 3.7 (3.7) |
| Refinement Statistics | |
| Resolution (Å) | 26.12–2.45 |
| No. reflections | 55,058 |
| $R_{work}/R_{free}$ | 0.21/0.26 |
| No. atoms | |
| Protein | 10,978 |
| Water | 335 |
| B-factors (Å$^2$) | |
| Protein | 57.52 |
| Water | 43.70 |
| RMS$^b$ deviations | |
| Bond lengths (Å) | 0.01 |
| Bond angles (°) | 0.84 |
| Ramachandran plot statistics | |
| % of favored region | 96.96 |
| % of allowed region | 3.04 |
| % of outlier region | 0 |

$^a$Values in parentheses refer to highest-resolution shell.
$^b$RMS root mean square.

X-ray crystallography, initial crystallization screening was performed on mature (signal sequence truncated) full length NADase/SLO and NADase$_{G330D}$/SLO complexes but failed to yield hits. To search for crystallizable constructs, various truncations of NADase and SLO were designed based on the knowledge of a partial structure of NADase in complex with its endogenous inhibitor IFS (PDB 3PNT and 4KT6)[35,36], and the structure of SLO (PDB 4HSC)[18]. Analysis of the elution profiles of size exclusion chromatography (SEC) revealed the dynamic N-terminal region of SLO (residues 35-105) is dispensable for binding to the C-terminal enzymatic domain of NADase (Supplementary Fig. 1). Diffraction-quality crystals of C-terminally His-tagged NADase$_{193-451,G330D}$ in complex with tag-free SLO$_{106-574}$ were produced, and the crystal structure of this complex was determined to a resolution of 2.45 Å by molecular replacement (Fig. 1a, b, and Table 1). Within the crystallographic asymmetric unit, two copies of the complex are present, with each complex consisting of NADase$_{193-451,G330D}$ and SLO$_{106-574}$ at a 1:1 molar ratio, consistent with the reported solution stoichiometry[17]. The overall structure of the NADase$_{193-451,G330D}$/SLO$_{106-574}$ heterodimer exhibits an elongated shape with the C-terminal domain 4 of SLO bound to the C-terminal enzymatic domain of NADase.

The structure of SLO-bound NADase$_{193-451,G330D}$ superimposes well with the structure of NADase$_{193-451}$ in complex with IFS (PDB 4KT6), with a root-mean-square deviation (RMSD) of 0.518 Å for 212 aligned C$_\alpha$ atoms, suggesting the common polymorphism G330D does not affect the global structure of the NADase enzymatic domain (Supplementary Fig. 2). Superimposing the structure of SLO in the NADase$_{193-451,G330D}$/SLO$_{106-574}$ complex onto the SLO-alone structure (PDB 4HSC), yields an RMSD of 1.942 Å for 424 C$_\alpha$ atoms, and reveals that domains 1 and 3 are well-aligned but the orientation of domain 2 differs slightly, and this is accompanied by a larger shift of domain 4 (Supplementary Fig. 2).

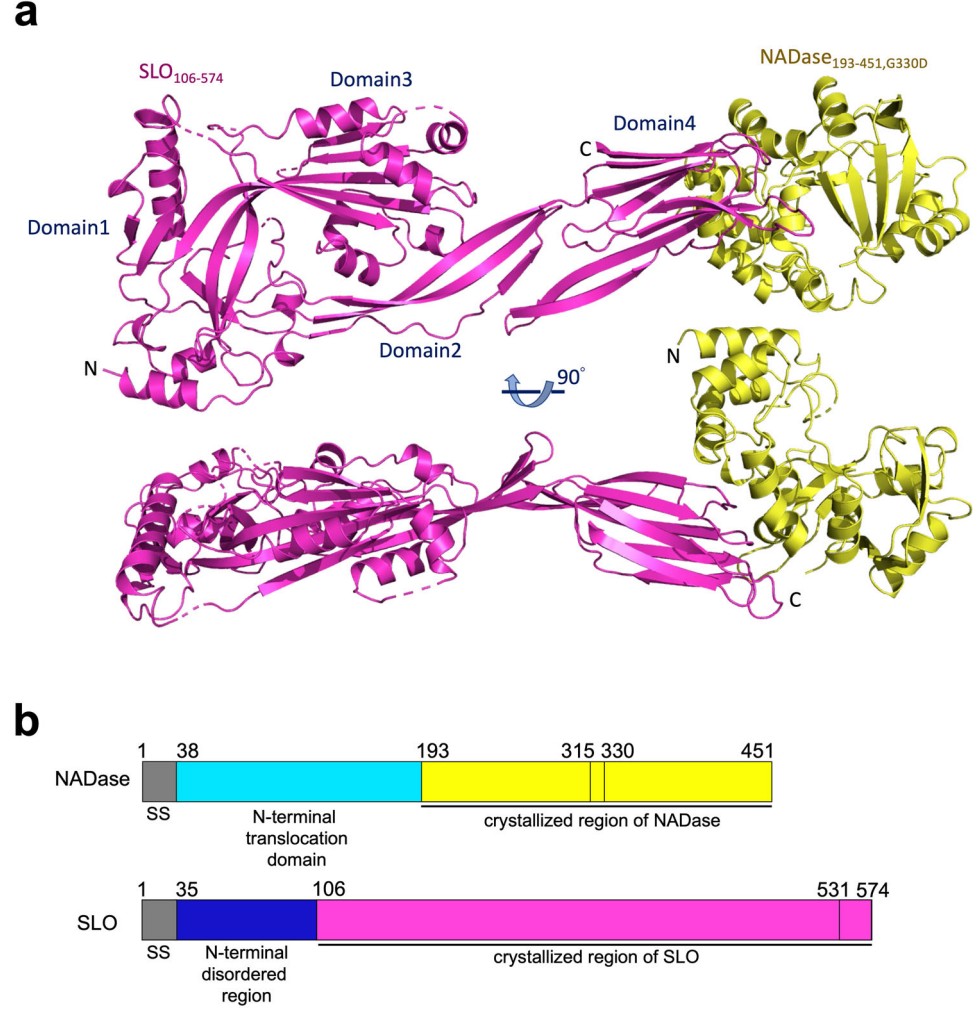

**Fig. 1 Crystal structure of the NADase$_{193-451,G330D}$/SLO$_{106-574}$ complex. a** Ribbon diagram of the crystal structure of the NADase$_{193-451,G330D}$/SLO$_{106-574}$ complex in two views related by a rotation of 90° about the horizontal axis. Yellow, NADase$_{193-451,G330D}$; Magenta, SLO$_{106-574}$. **b** Schematic diagrams of NADase and SLO: Grey, secretory signal (SS); Cyan, N-terminal translocation domain of NADase; Yellow, C-terminal enzymatic domain (crystallized region); Blue, N-terminal disordered/dynamic region of SLO; Magenta, domains 1 to 4 of SLO of NADase (crystallized region). The NADase mutant G330D lacks detectable NADase activity, while a key salt bridge is observed between NADase residue D315 and SLO residue R531.

In the tail-to-tail interface of the NADase$_{193-451,G330D}$/SLO$_{106-574}$ heterodimeric structure, 13 residues from loop1, loop2, loop3 and the undecapeptide of SLO domain 4, make contacts with 13 residues from helices α2, α4, and α6 at the C-terminus of NADase. These interactions are mediated by numerous hydrogen bonds and one electrostatic interaction (Fig. 2a and Supplementary Fig. 3). The buried surface area of the complex is ~863 Å$^2$, which is at the smaller end of the range of buried surface area measures of typical protein complexes[37]. Among the interacting residues, NADase residue D315 forms the only salt bridge with SLO residue R531. To validate the interface interaction, we generated a point mutation on NADase that replaces the negatively charged residue D315 with a positively charged arginine residue by site-direct mutagenesis. The recombinant mutant, NADase$_{D315R}$, was mixed with SLO and subjected to SEC for analysis of complex formation. The elution profile shows NADase$_{D315R}$ did not co-elute with SLO (Fig. 2b), indicating the disruption of complex formation. To verify the dissociation of NADase/SLO complex was due to the salt-bridge disruption rather than a major structural change of NADase caused by the point mutation, the NADase activity assay was performed. NADase$_{D315R}$ exhibited comparable activity to the wild-type NADase in cleaving β-NAD$^+$ (Supplementary Fig. 4), indicating that the overall structures of NADase$_{D315R}$ and wild-

type NADase are highly similar. To further confirm the key role of the salt bridge in complex formation, a complementary mutation (R531D) in SLO was constructed to test whether SLO$_{R531D}$ could restore the electrostatic interaction with NADase$_{D315R}$. The SEC profile showed NADase$_{D315R}$ and SLO$_{R531D}$ co-eluted in a single peak at the same elution volume as the wild-type NADase/SLO complex (Fig. 2b), indicating NADase$_{D315R}$/SLO$_{R531D}$ restores the ionic interaction required for complex formation. These results demonstrate the salt-bridge pair between NADase D315 and SLO R531 plays a key role in the interface interaction.

**Small-angle scattering studies on the NADase/SLO complex.** The N-terminal translocation domain of NADase and the N-terminal disordered region of SLO are both essential for SLO-mediated translocation of NADase across the host cell membrane via CMT[38,39]. However, these parts are missing in the determined crystal structure of the NADase$_{193-451,G330D}$/SLO$_{106-574}$ complex. To gain structural insights into the missing structural elements required for CMT activity, we conducted small-angle X-ray and neutron scattering (SAXS and SANS) on mature full-length NADase and the NADase/SLO complex. SEC coupled with SAXS (SEC-SAXS) was performed to ensure the scattering data were

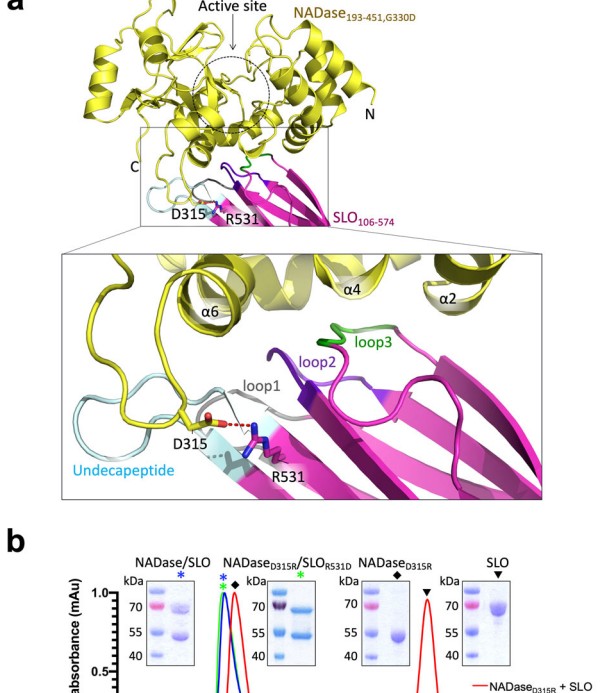

**Fig. 2 The NADase$_{193-451,G330D}$/SLO$_{106-574}$ interface. a** Loop1 (grey), loop2 (purple), loop3 (green), and undecapeptide (light blue) in the C-terminal domain 4 of SLO interacts with α2, α4, and α6 of NADase C-terminus (yellow). A salt bridge (red dash) is formed between NADase D315 and SLO R531 residues shown as sticks. For simplicity, hydrogen-bonding interactions are shown in Supplementary Fig. 3. **b** Size exclusion chromatographic analysis of complex formation. The elution profile and SDS-PAGE show the NADase D315R mutant disrupted the NADase/SLO complex, and NADase$_{D315R}$/SLO$_{R531D}$ restored complex formation. Note that the elution peak of SLO was delayed, consistent with the previous study[17].

free of aggregation. SAXS data collected from NADase and the NADase/SLO complex showed a linear behavior in the Guinier region consistent with a monodisperse solution of macromolecules (Fig. 3a; Supplementary Fig. 5a, b and Supplementary Table 1). The pair-distance distribution function $P(r)$ calculated from the NADase SAXS data shows a major peak with the prominent shoulder, indicating a well-separated two-domain system with a maximum dimension $D_{max}$ of ~103 Å (Fig. 3b). The $P(r)$ of NADase/SLO showed a prolonged tail with the $D_{max}$ of ~184 Å, suggesting the presence of extended conformations in solution (Fig. 3b). Ab initio molecular envelopes were generated to depict the overall architecture of NADase and NADase/SLO complex (Fig. 3c).

To further understand the structural details of how the N-terminal translocation domain of NADase, which is missing in our determined crystal structure, interacts with SLO, we performed SAXS-based atomistic modeling of the full-length NADase and NADase/SLO complex. Currently, only partial atomic structures of NADase are available: the C-terminal domain in complex with IFS[35,36] and a recently determined structure of the N-terminal domain by nuclear magnetic resonance[40]. To delineate how the domains are linked, a structural model of full-length NADase was generated by AlphaFold[41]. To validate the model, the theoretical scattering

profile was computed from the single model of NADase. However, it showed a poor fit to the SAXS curve ($\chi^2 = 17.8$, Fig. 3a). The dimensionless Kratky plot of NADase shows the peak shift to the right with the maxima >1.104, suggesting interdomain flexibility[42] (Supplementary Fig. 5c). Thus, conformational sampling was performed using BILBOMD[43], followed by the selection of a multistate model of NADase that best fits SAXS data[44]. The multistate model of NADase with a flexible N-terminal domain matches the SAXS data significantly better than a single conformer ($\chi^2 = 1.9$, Fig. 3a, c). The full-length NADase was combined with the full-length SLO model to build an initial full-length model of NADase/SLO complex for further SAXS-based modeling. The dimensionless Kratky plot of NADase/SLO complex also showed a peak shift to the right with a maxima >1.104 suggestive of flexibility (Supplementary Fig. 5c). Thus, the conformational space sampled by both the N-terminal disordered region of SLO and the flexible domain of NADase in complex was further explored by BILBOMD. A multistate model with 9% of compact and 53% of extended state, together with 38% of free NADase, fits well with the SAXS data and comprehensively describes the solution state of NADase/SLO complex (Fig. 3a, c). The presence of free NADase in the structural ensemble is attributed to the known behavior of partial dissociation of NADase/SLO complex in solution[17]. The dissociated SLO displayed a behavior of delayed elution in SEC[17] (Fig. 2b) and thus did not contribute to the SAXS signal of NADase/SLO in SEC-SAXS. The dissociation of NADase/SLO was confirmed by multi-angle light scattering coupled with SEC (SEC-MALS) (Supplementary Fig. 6) that showed the molecular weight of the NADase/SLO complex is ~90 kDa, lower than the expected mass of ~110 kDa and consistent with a previous report[17]. In addition to SAXS studies, we attempted to visualize each protein within the NADase/SLO complex by SANS with contrast variation, where the measurements were designed such that SLO would be contrast matched in 42% $D_2O$ and the deuterium labelled NADase would be contrast matched in 100% $D_2O$. The latter measurement was of particular importance as it would yield direct insight into the conformation of SLO while bound to NADase. Unfortunately, the 100% $D_2O$ sample showed signs of aggregation/oligomerization, which made interpretation of this contrast point infeasible. Nonetheless, the other SANS contrast points measured from the deuterated-NADase/SLO complex were successfully analyzed in terms of the same multistate model used to fit the SAXS data, and it was found that the contrast variation data was in excellent agreement with the same relative proportions of extended and compact complex, plus free NADase (Supplementary Fig. 7 and Supplementary Table 2). Thus, the SANS provides additional support that the NADase/SLO complex exhibits dynamic behavior, involving a rigid NADase/SLO core structure with dynamic peripheral domains.

The multistate ensemble of the NADase/SLO complex shows two distinct conformations adopted by the SLO N-terminal disordered region and the NADase N-terminal domain. The translocation domain of NADase makes contacts with SLO domain 4 in the representative compact state (Fig. 3c lower left), but does not interact with SLO in the representative extended state (Fig. 3c lower right). This structural ensemble of the NADase/SLO complex indicates why the N-terminal translocation domain is dispensable for the complex formation but is required for the maximal binding of NADase to SLO[17] and for the SLO-mediated translocation of NADase. Specifically, the N-terminal dynamic region of SLO and the N-terminal domain of NADase do not interact in the representative extended state (Fig. 3c lower right), resulting in less extensive contacts relative to the compact state. As the dissociation process of a protein complex often occurs through a number of transition events

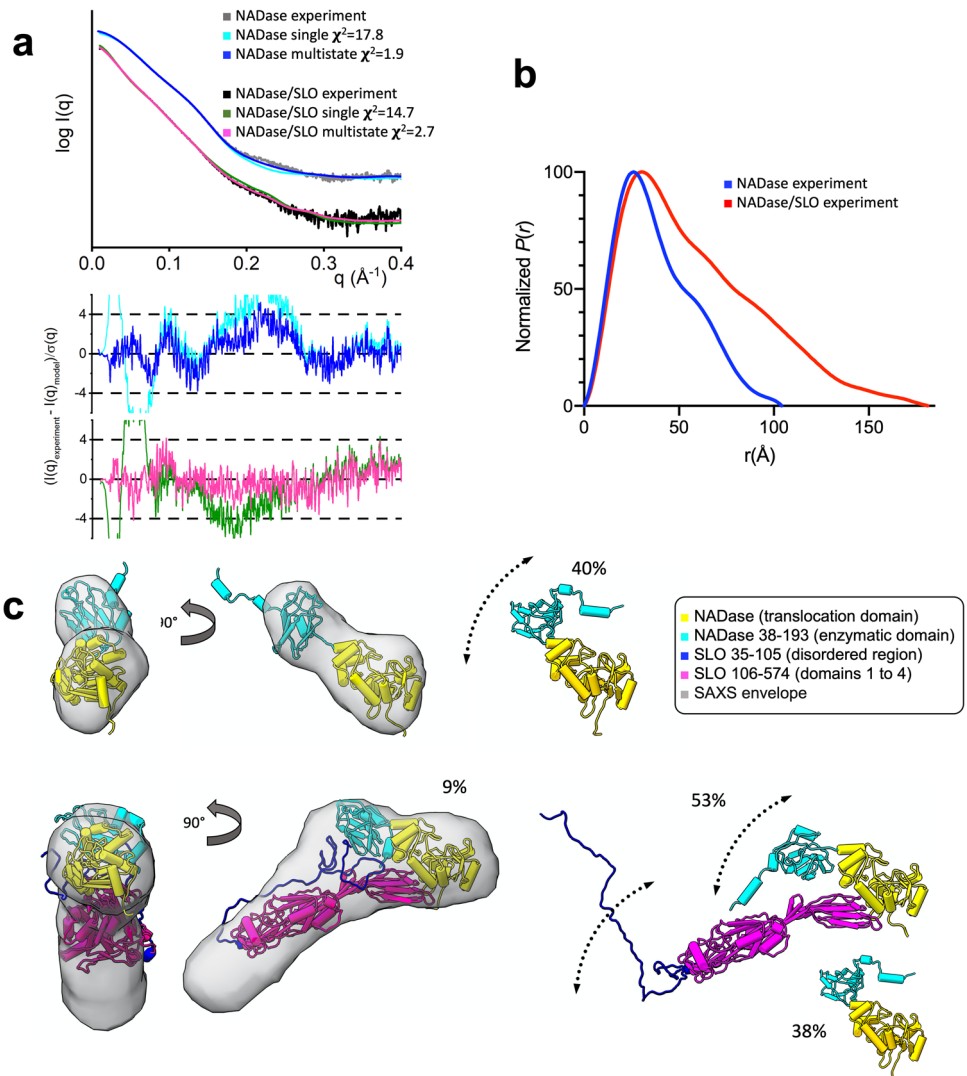

**Fig. 3 Conformational ensemble of NADase and NADase/SLO complex by SAXS analysis. a** Experimental curves of NADase and NADase/SLO along with the theoretical scattering profiles and fit-residuals for single best (NADase, cyan; NADase/SLO, dark green) and multistate models (NADase, blue; NADase/SLO, pink). The multistate models match the SAXS data significantly better than a single conformer for both NADase ($\chi^2 = 1.9$ vs. 17.8) and NADase/SLO ($\chi^2 = 2.7$ vs. 14.7). **b** Distance distribution function [$P(r)$] of NADase (blue) and NADase/SLO complex (red). **c** Multistate models of NADase and the NADase/SLO complex. One conformer of NADase and NADase/SLO is superimposed on the average SAXS molecular envelope and shown in two orthogonal views.

involving low surface area contact interfaces[45], the compact-to-extended state transition could be interpreted as the first step of the dissociation process of the NADase/SLO complex. This may suggest that the dynamic motions of the N-terminal domain of NADase and the N-terminal disordered region of SLO are functionally linked to the CMT activity which involves the separation of NADase from SLO.

Mapping the polymorphic residues of NADase identified from clinical isolates[34] on the full-length structural model of the NADase/SLO complex reveals the polymorphisms are distant from the complex interface (Supplementary Fig. 8a), supporting the notion that the preservation of the NADase/SLO complex plays an important role in the molecular evolution of NADase[46]. Moreover, a recent report shows NADase residues, Tyr120, Tyr121, Tyr151, and His177, are involved in mediating carbohydrate recognition and membrane binding[40]. These residues are located at the solvent-exposed area in the NADase/SLO multistate model (Supplementary Fig. 8b), showing the structural ensemble of the NADase/SLO complex is competent

for binding to cell surface glycans and is functionally relevant. Taken together, the small-angle scattering studies reveal the dynamic nature and functional conformations of the NADase/SLO complex, and provide molecular insight into the evolution of NADase.

**GAS genome harboring an NADase/SLO complex-defective mutation impairs SLO-mediated translocation of NADase and resistance to phagocytic killing in vitro.** It has been demonstrated that in GAS, deletion of SLO, NADase, or both compromises the ability of bacteria to induce cytotoxicity in host cells and to survive within immune cells[14,15,21,23]. However, the effect of disrupting specific interactions between NADase and SLO on the pathogenic abilities of GAS has not been tested, and can only be tested in an informed manner when structural information is available. The determined structure in this study allows us to pinpoint and mutate key interacting residues in the GAS genome to assess the importance of specific NADase/SLO interactions in the pathogenicity of GAS. Thus, GAS mutants engineered to

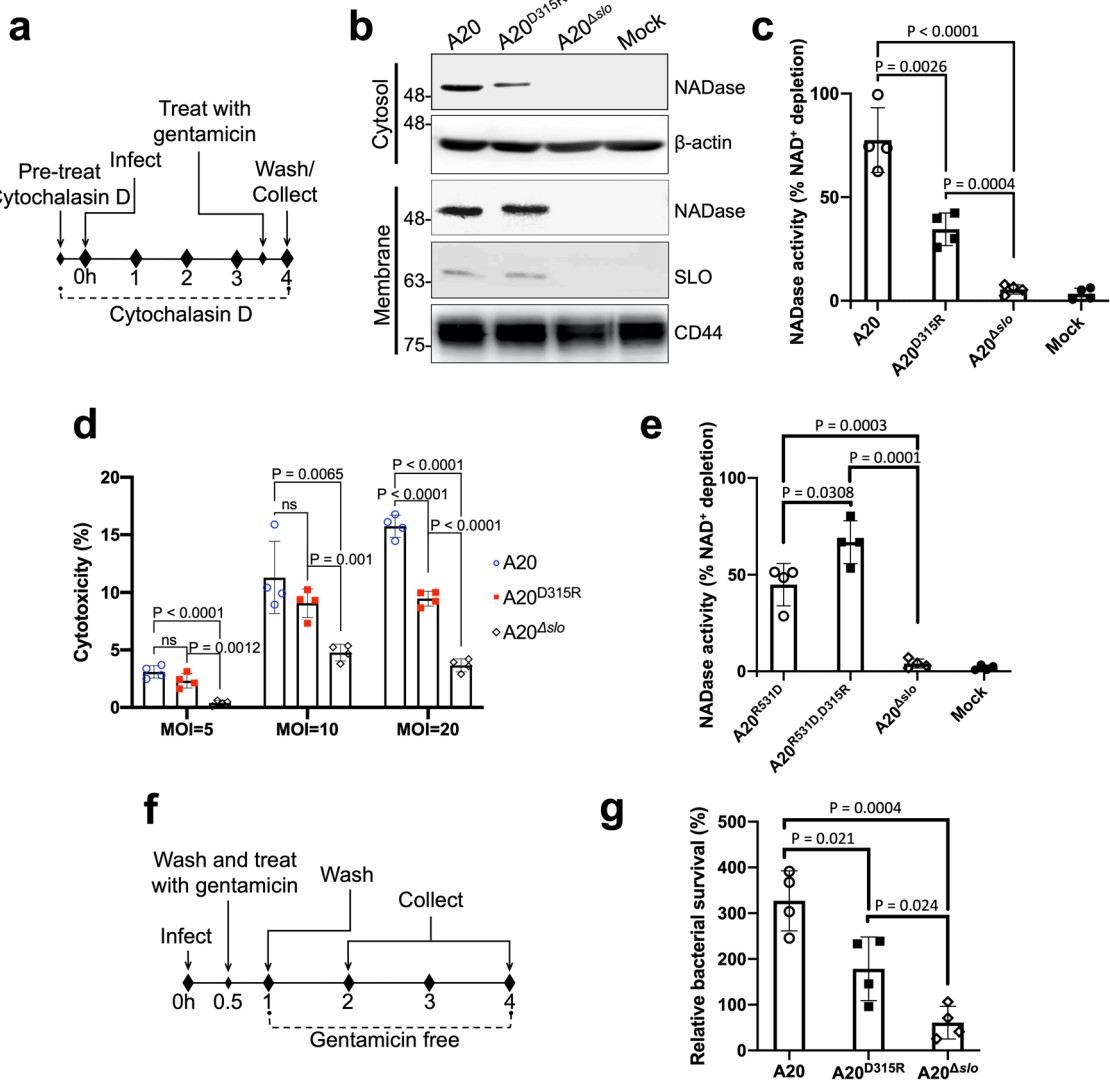

**Fig. 4 The impacts of the NADase-SLO interaction on the CMT-mediated cytotoxicity of GAS and the bacterial survival within immune cells. a** The schematic protocol for analyzing CMT-mediated translocation of NADase. A549 cells were pretreated with the microfilament inhibitor cytochalasin D to block bacterial entry[26], followed by infection with A20, A20$^{D315R}$, or A20$^{\Delta slo}$ at an MOI of 5. **b** Immunoblot analyses of cytosolic (intracellular) and membrane fractions harvested from the GAS-infected A549 cells. Immunoblots were performed with antibodies against the proteins indicated on the right. β-actin (cytosol) and CD44 (membrane) were included as loading controls. **c** Intracellular NADase activity. The levels of NADase translocated into A549 cells were quantified by measuring cytosolic (intracellular) NADase activity of the GAS-infected A549 cells. **d** Cytotoxicity of GAS strains to A549 cells. The cytotoxicities of A20, A20$^{D315R}$, or A20$^{\Delta slo}$ on A549 cells were determined by measuring the activities of lactate dehydrogenase released from the cells infected by the strains for 4 h at indicated MOIs. Cytotoxicities are expressed as a percentage of lactate dehydrogenase released by uninfected cells lysed with 1% Triton X-100. **e** The effect of restoring the NADase-SLO interaction on NADase translocation. Intracellular NADase activities of A20$^{R531D}$- and A20$^{R531D,D315R}$-infected A549 cells were measured after the infection protocol described in Fig. 4a was performed. **f** Schematic protocol of GAS infection of phagocytic cells. PMA-activated U937 cells were infected with A20, A20$^{D315R}$, or A20$^{\Delta slo}$ at an MOI of 50. **g** Survival of the GAS strains in U937 cells. The number of surviving bacteria was counted by a CFU-based assay. The results are presented as percentages of the intracellular bacterial burden at 4-hour post-infection compared to those at 2-hour post-infection. The data are the representative of three biologically independent experiments performed in quadruplicate shown as mean ± standard deviation. The statistical significance was calculated by a two-tailed unpaired t-test.

disrupt salt bridge formation between NADase D315 and SLO R531 were generated to assess the importance of this specific interaction to GAS cytotoxicity and GAS survival within immune cells.

To evaluate the impact of the NADase-SLO interaction on GAS cytotoxicity, we first assessed the contribution of the salt-bridge to NADase translocation into host cells during bacterium-host cell interaction via SLO-mediated translocation known as CMT[26]; an essential step for GAS to exert its full cytotoxic potential. As shown in Fig. 4a, A549 epithelial cells were incubated with GAS strains A20, A20$^{D315R}$, or A20$^{\Delta slo}$ at a multiplicity of infection

(MOI) of 5 for assessing the translocation of NADase. A20 expresses wild-type NADase and SLO, while A20$^{D315R}$ expresses the NADase D315R mutant and wild-type SLO. A20$^{\Delta slo}$ is an *slo* deletion mutant of A20 and thus does not express SLO. After 4 h of incubation, the intracellular NADase levels in A20$^{D315R}$-infected cells were significantly lower than in A20-infected cells and were undetectable in the A20$^{\Delta slo}$-infected cultures (Fig. 4b). The corresponding intracellular NADase activity in A20$^{D315R}$-infected cells was significantly lower than that in A20-infected cells, while no activity was detected in A20$^{\Delta slo}$-infected cells (Fig. 4c). In addition, the lower intracellular NADase protein

levels and activities in A20$^{D315R}$-infected cells were not due to a reduced secretion level or a decreased activity of A20$^{D315R}$ strain (Supplementary Fig. 9a–c), confirming the introduction of the complex-defective mutation in the GAS genome impairs CMT activity resulting in a lower level of NADase translocation. Next, the cytotoxic effect associated with intracellular NADase was inspected. Small differences in NADase-induced cytotoxicity between A20 and A20$^{D315R}$ were detected at the initial MOI of 5 (Fig. 4d). To test whether a higher MOI treatment would result in a more significant difference between the A20- and A20$^{D315R}$- induced cytotoxicity, MOIs were increased to 10 and 20. Statistically significant differences in NADase-mediated cytotoxicity between A20 and A20$^{D315R}$ were observed at an MOI of 20 (Fig. 4d). These data demonstrate the disruption of the salt bridge formation between NADase and SLO in A20$^{D315R}$ impairs the functional delivery of NADase into the host cytosol via CMT and thus results in weaker cytotoxic activity. Moreover, given that SLO$_{R531D}$ could rescue the electrostatic interaction with NADase$_{D315R}$ (Fig. 2b), we investigated whether introducing SLO$_{R531D}$ into the GAS mutant that expresses NADase$_{D315R}$ can restore the ability in NADase translocation. For this purpose, we constructed two A20 mutants: A20$^{R531D}$ which expresses SLO$_{R531D}$ and wild-type NADase, and A20$^{R531D,D315R}$ which expresses SLO$_{R531D}$ and NADase$_{D315R}$. To our surprise, significantly lower levels of SLO protein secretion were observed in A20$^{R531D}$ and A20$^{R531D,D315R}$ compared to A20 and A20$^{D315R}$ (Supplementary Fig. 9d, e), suggesting the replacement of arginine residue with aspartic acid at position 531 of SLO decreased the secretion level of SLO. Given that the translocation of NADase is SLO-dependent, the discrepancy in the SLO secretion levels between A20 and A20$^{R531D,D315R}$ did not allow us to properly evaluate whether rescuing the salt bridge could restore the ability of NADase translocation in A20$^{R531D,D315R}$. On the other hand, A20$^{R531D}$ and A20$^{R531D,D315R}$ showed comparable levels of SLO secretion (Supplementary Fig. 9d, e) permitting evaluation of the role of salt bridge formation in NADase translocation. A20$^{R531D,D315R}$ exhibited a higher level of NADase translocation than A20$^{R531D}$ during GAS-A549 infection (Fig. 4e), demonstrating the salt bridge at the NADase/SLO complex interface plays a key role in NADase translocation.

To study the impact of the complex-defective mutation on GAS resistance to phagocytic killing, differentiated U937 cells were infected with A20, A20$^{D315R}$, or A20$^{Δslo}$ at an MOI of 50, and the bacterial survival inside phagocytic cells was assessed (Fig. 4f). Compared to A20 at 4 h of post-infection, A20$^{D315R}$ exhibited reduced survivability within U937 cells, while A20$^{Δslo}$ showed the lowest survival (Fig. 4g). These findings suggested that the NADase-SLO interaction contributes to the survival within phagocytes. The same experimental setting also found that A20 and A20$^{D315R}$ induced similar levels of cytotoxicity on U937 cells (Supplementary Fig. 10), suggesting the difference in intracellular survival between the two strains is not caused by their cytotoxic effects.

**A specific NADase-SLO interaction contributes to the pathogenesis of GAS in a mouse infection model**. To further assess the biological importance of the salt bridge interaction for NADase/SLO complex formation in the pathogenesis of GAS during infections, mice were subjected to an air-pouch model of subcutaneous infection[47–49] with A20 or A20$^{D315R}$ ($n = 8$ for each GAS strain). At 24-hour post-infection, A20$^{D315R}$ induced significantly smaller skin lesions than A20 (Fig. 5a, b and Supplementary Fig. 11). The A20$^{D315R}$-infected animals exhibited lower bacterial burdens in the infected tissues than the A20-infected ones (Fig. 5c). Consistently, compared to A20-infected

mice, A20$^{D315R}$-infected animals produced lower levels of the inflammatory cytokine IL-1β (Fig. 5d), which is known to be stimulated by GAS infections[48,50]. These findings showed the NADase-SLO salt-bridge interaction contributes to the virulence of GAS during infections. Overall, the results from this mouse infection study reveals the significance of the ionic interaction at the NADase/SLO complex interface in GAS pathogenesis.

**Discussion**

NADase-SLO synergism utilized by GAS serves as a unique example that two secreted toxins of Gram-positive bacteria interact and act in a concerted manner. The coordinated actions of NADase and SLO in increasing cytotoxicity[16,17,26], enhancing invasiveness[5–7,11], promoting intracellular survival[14,15,28–31], and mediating immune evasion[14,15,21,23] have been well-documented, illustrating the uniqueness and specificity of GAS pathogenesis. Prior to this work, it was not understood how NADase and SLO assemble into a complex to accomplish these tasks and how NADase-SLO interactions correlate with their synergistic functions. Here, we uncover the molecular structure of the NADase/ SLO complex, revealing that a salt bridge formed between NADase D315 and SLO R531 is fundamental for the interface interaction and that conformational dynamics play an important role to the functioning of the complex. Furthermore, we demonstrate that disruption of the NADase-SLO interaction can affect the outcomes of GAS infection in vitro and in our mouse model. Our studies convey the structure-functional relationship of the NADase/SLO complex underpinning the augmented virulence of GAS.

For over a decade, the biological relevance of the NADase-SLO interaction in GAS pathogenesis could only be demonstrated by deletion of either toxin or both[5,7,11,14,15] due to the lack of structural information. The atomic details of the NADase$_{193-451,G330D}$/SLO$_{106-574}$ complex determined by X-ray crystallography in this study (Figs. 1 and 2) allow us to design experiments to evaluate the importance of specific NADase-SLO interactions in GAS virulence by introducing point mutations in the GAS genome rather than gene deletion. In this study, we demonstrate that disrupting the interaction between these two toxins impairs GAS cytotoxicity and intracellular survival (Fig. 4). Furthermore, the significance of NADase/SLO complex formation on the pathogenesis of GAS infection is directly demonstrated in the murine model of skin infection (Fig. 5).

Structural characterization of the full-length NADase/SLO complex by combined use of X-ray crystallography, computational modeling, and small-angle scattering reveals an ensemble of a compact and an extended states (Fig. 3). The relative mobility of the N- and C-terminal domains of NADase and the large dynamic motion of the SLO N-terminus add to the complexity to the NADase-SLO interaction. The compact state displays a physical association between the NADase N-terminal domain and SLO and this larger contact surface area provides a structural basis for strong binding of NADase to SLO[17]. The extended state, with its a lower contact surface area at the NADase/SLO interface, is suggestive of a transient conformation during the dissociation process that could be of importance during NADase translocation. Additionally, structural analyses on the full-length atomistic models of the NADase/SLO complex suggest that both compact and extended states are competent in NADase-mediated glycan-binding (Supplementary Fig. 8b) and that the interface interaction is conserved in the molecular evolution of NADase (Supplementary Fig. 8a). Overall, SAXS studies reveal functionally-relevant conformations of the NADase/SLO complex and suggest the dynamic interplay between SLO and NADase is fundamental to the functioning of the complex and the molecular clues

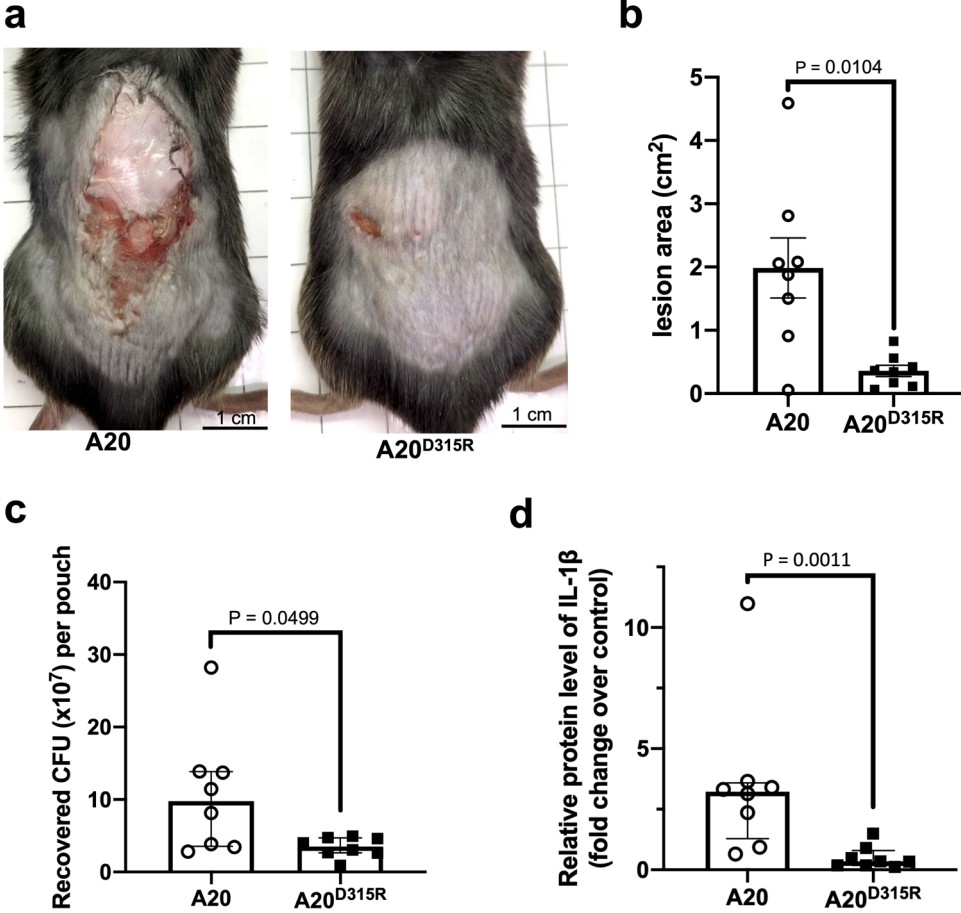

**Fig. 5 GAS-infected mouse model. a** Representative images of skin lesions of A20- or A20$^{D315R}$-infected mice at 24-hour post-infection. Scale bars: 1 cm. **b** Quantification of the skin lesion sizes. The area of necrotic skin lesions at 24-hour post-infection was quantified by analyzing the lesion images. There were 8 mice ($n = 8$) used for each infection group. **c** Bacterial counts in air pouch exudates. The bacterial numbers of A20 and A20$^{D315R}$ were counted by a CFU-based assay. **d** Quantification of IL-1β. The levels of IL-1β in the GAS-infected skin tissues are presented as the ratio of the level of IL-1β related to that of β-actin. The quantitative results are presented as median with interquartile range ($n = 8$). The statistical significance was evaluated by a two-tailed Mann-Whitney $U$ test.

provided here will be beneficial for future studies to dissect the mechanistic actions of NADase and SLO in CMT.

Taken together, this work delivers structural insight into the synergism of NADase and SLO in GAS pathogenesis and fills a current knowledge gap in the literature. Currently, vaccine development remains challenging due to the diversity of GAS strains and antigenic variation[51]. NADase has been considered as a therapeutic target for treating GAS infection[52]; however, interest has subsided as NADase enzymatic deficient variants found in clinical isolates remain pathogenic. Our structural information presented here may open up an opportunity for the design of alternative agents to perturb the NADase-SLO interaction to suppress GAS virulence.

## Methods

**Plasmid construction, protein expression and purification.** Genes from GAS strain A20 genomic DNA encoding NADase (accession number: AFV37283) and SLO (accession number: AFV37285) and were amplified by PCR and cloned into vectors pET-43.1a and pET-28a, respectively. The constructs subsequently served as templates for PCR using appropriate primers to generate constructs encoding the C-terminally His-tagged NADase proteins (NADase$_{G330D}$, NADase$_{D315R}$, and NADase$_{193-451,G330D}$), N-terminally His-tagged SLO proteins (SLO and SLO$_{R531D}$), and tag-free SLO$_{106-574}$. The N-termini of mature full length NADase and SLO were residues 38 and 35 respectively, as indicated in Fig. 1. A spontaneous point mutation for NADase$_{G330D}$, G368R, was observed in our clone which improved the solubility of recombinant NADase$_{G330D}$. NADase and its variants were expressed in *Escherichia coli* Rosetta (DE3). N-terminally His-tagged SLO

and tag-free SLO$_{106-574}$ were expressed in *E. coli* BL21 (DE3); Expression of the recombinant proteins (NADase$_{G330D}$, NADase$_{D315R}$, and NADase$_{193-451,G330D}$, SLO, SLO$_{106-574}$, and SLO$_{R531D}$) were induced by IPTG (isopropyl-β-D-thiogalactopyranoside) when cells reached to an O.D.$_{600}$ of 0.4–0.6, and further incubated at 16 °C for 18 h. Cell pellets were harvested, resuspended in buffer (20 mM HEPES pH7.5, 500 mM NaCl, 10% glycerol, 1 mM β-mercaptoethanol), and disrupted by sonication. Supernatant was loaded onto a cobalt affinity column (TALON superflow, GE Healthcare) and unbound proteins were washed with 30 mM imidazole in buffer A (20 mM HEPES pH 7.5, 300 mM NaCl, 1 mM β-mercaptoethanol). Proteins were eluted with 300 mM imidazole in buffer A. Fractions containing proteins were pooled and further purified by SEC (HiLoad 16/600 Superdex 200 prep grade, GE Healthcare). For SANS, deuterated C-terminally His-tagged NADase$_{G330D}$ was produced at the National Deuteration Facility, Australian Nuclear Science and Technology Organisation using 100% D$_2$O, and 40 g/l $^1$H glycerol as the sole carbon source[53]. The tag-free NADase and NADase$_{D315R}$ were expressed and purified[29]. To reconstitute NADase/SLO, purified tag-free NADase and N-terminally His-tagged SLO with a concentration of 25 to 30 μM were mixed at a molar ratio of 1:1 in phosphate buffered saline (PBS) buffer pH 7.4 with 1 mM DTT, and subjected to SEC. For NADase$_{193-451,G330D}$/SLO$_{106-574}$ complex preparation, the bacterial cultures containing 6xHis-tagged NADase$_{193-451,G330D}$ and tag-free SLO$_{106-574}$ were mixed and disrupted by sonication on ice. Supernatant was loaded into a cobalt affinity column (TALON superflow, GE Healthcare) and unbound proteins were washed with 30 mM imidazole in buffer A. NADase$_{193-451,G330D}$/SLO$_{106-574}$ complex was eluted with 300 mM imidazole in buffer A. Fractions containing NADase$_{193-451,G330D}$/SLO$_{106-574}$ complex were further purified by SEC (HiLoad 16/600 Superdex 200 prep grade, GE Healthcare).

**Crystallization, crystallographic data collection and structural determination.** Crystals of the NADase$_{193-451,G330D}$/SLO$_{106-574}$ complex were grown at 295 K by

vapor diffusion in sitting drops composed of equal volumes of protein solution (12 mg/ml in 10 mM HEPES pH 7.4, 150 mM NaCl, 1 mM TCEP, and 1% glycerol) and reservoir solution (50 mM HEPES pH 7.3 and 17.5% PEG 3350). All crystals were cryoprotected by soaking in crystallization solution supplemented with 15% (v/v) glycerol for 10 s and then flash-cooled to 100 K in liquid nitrogen. Crystal-lographic datasets were collected on beamlines TLS13B1, TLS15A1, and TPS05A (wavelength = 1 Å, temperature = 100 K) at the National Synchrotron Radiation Research Center in Hsinchu, Taiwan. Diffraction data were processed by HKL2000[54]. The crystal structure of NADase$_{193-451,G330D}$/SLO$_{106-574}$ complex was solved by molecular replacement performed with PHASER[55] using the structures of SLO (PDB 4HSC) and NADase in NADase/IFS complex (PDB 4KT6) as search models. The atomic model was built by COOT[56]. Refinement using program PHENIX[57] was iteratively alternated with model building. Missing residues were not built due to weak electron densities. The final model includes NADase$_{193-451,G330D}$ residues 194–366 and 368–445, and SLO$_{106-574}$ residues 106–170, 172–174, 176–247, 249–376, 381–398, 404–427 and 429–574. Crystallographic data and refinement statistics are shown in Table 1. Coordinates and structure factors with the identifier have been deposited in protein data bank with code 7WVH.

**Enzymatic assay of recombinant NADase.** The enzymatic activities of the recombinant NADase and NADase$_{D315R}$ were assessed by hydrolyzing β-NAD$^+$ into nicotinamide and ADP-ribose[29]. Recombinant NADase and NADase$_{D315R}$ were two-fold serially diluted to 15–20 nM in PBS. Each sample was incubated with 0.67 mM β-NAD$^+$ (Sigma–Aldrich) at 37 °C for 1 h. The reaction was terminated by the addition of 2 M NaOH. Following an additional 1-hour incubation at room temperature in the dark, the level of uncleaved β-NAD$^+$ in each reaction was determined by an ELISA reader (360 nm absorbance; Multiskan FC, Thermo). The NADase activity of each sample was presented as the percentage of NAD$^+$ consumption relative to the PBS buffer-only control.

**SAXS data collection and analysis.** Data were collected at the SIBYLS beamline 12.3.1[58], Advanced Light Source, Lawrence Berkeley National Laboratory, Berkeley, CA, USA at 1.03 Å wavelength with a Pilatus 2 M detector at 1.5 m sample-to-detector distance, corresponding to a q range from 0.01 to 0.4 Å$^{-1}$. The scattering vector is defined as $q = 4\pi \sin\theta/\lambda$, where $2\theta$ is the scattering angle, and λ is the X-ray wavelength. NADase (12.3 mg/ml) and NADase/SLO complex (15.1 mg/ml) were subjected to SEC-SAXS[59]. The SEC column (Shodex KW-803) was equilibrated with PBS buffer supplemented with 1 mM DTT at a flow rate of 0.5 ml/min. A total of 50 µl of each sample was injected onto the SEC column and 3-second X-ray exposures were collected continuously during the elution. The SAXS frames recorded prior to the protein elution peak were taken to be the solvent scattering and were subtracted from all other frames. The elution peak was mapped by comparing integral ratios to background and radius of gyration $R_g$ relative to the recorded frame using the program SCÅTTER. Final merged SAXS profiles were derived by integrating multiple frames across the elution peak that were deemed equivalent, and these merged data were used for further analysis. The merged SAXS curve was investigated by Guinier approximation $I(q) = I(0)\exp(-q^2 R_g^2/3)$ with the limits $q.R_g < 1.1$ that also determined radius of gyration ($R_g$). (Supplementary Fig. 5). The program GNOM[60] was used to compute the pair distribution function ($P(r)$) and to estimate the distance $r$ where $P(r)$ approaches zero, which corresponds to the macromolecule's maximal dimension ($D_{max}$). $P(r)$ functions were normalized at the maxima.

**SAXS modeling.** A low-resolution ab initio molecular envelope was computed using the program GASBOR[61] and the resulting reconstructions were averaged and filtered by the program DAMAVER[62]. For atomistic modeling, the structural model of NADase generated by AlphaFold[41] and the crystal structure of NADase$_{193-451,G330D}$/SLO$_{106-574}$ complex (PDB 7WVH) were used as initial models. The missing loops and linkers in the crystal structure were built with the program MODELLER[63]. The full-atomic models were used as templates for rigid-body modeling in the program BILBOMD[43], which applies the molecular dynamics simulation to explore conformational space of the flexible domains. In the BILBOMD strategy, the theoretical SAXS profile for each registered conformation and the corresponding fit to the experimental data were calculated using the program FoXS[64]. MultiFoXS[44] was performed to select appropriate multistate model of NADase and, NADase/SLO with free NADase in solution. The best multistate model was determined by minimizing the discrepancy χ$^2$ between the experimental scattering intensity data $I_{exp}(q)$ and the calculated multistate SAXS profile $I_{mod}(q)$.

**SEC-MALS.** Molecular masses were determined using a SEC column (Shodex KW-803) coupled with Dawn HELEOS II MALS and Optilab T-rEX Refractive Index Detector (Wyatt Technology). The system was equilibrated with PBS buffer supplemented with 1 mM DTT. A total of 50 µl of each sample was injected onto the SEC column at a flow rate of 0.5 ml/minute. Data were analyzed using ASTRA 6 software (Wyatt Technology). Bovine serum albumin was used to calibrate and normalize the detectors.

**SANS data collection and analysis.** SANS data were collected on the QUOKKA instrument[65] at the Australian Nuclear Science and Technology Organisation (Supplementary Table 2). Two aliquots of deuterated-NADase/SLO were obtained from SEC using a Superdex 200 increase 10/300 column. The first aliquot was obtained from peak fractions of the protein run in a buffer containing H$_2$O (the 0% D$_2$O sample), while the second aliquot was obtained from peak fraction of the protein run in a buffer containing D$_2$O (the 100% D$_2$O sample). The concentration of each aliquot was adjusted to 3.5 mg/ml assessed by the absorbance of the sample at 280 nm and correcting for the calculated extinction coefficient of the complex. Samples containing 20%, 42%, and 60% D$_2$O were obtained by mixing the two original aliquots at the appropriate ratios. The two-dimensional data were normalized by the incident neutron beam intensity and corrected for sample transmission, background radiation, empty cell scattering, detector sensitivity and radially averaged to produce $I(q)$ vs. $q$ profiles. Scattering data from the two different instrument settings were then merged, and buffer scattering data were then subtracted from the protein + buffer data to give the protein scattering profiles (the 20%, 42%, and 60% buffer scattering curves were taken as a linear combination of the 0% and 100% buffer scattering curves). To correct for the effects of incoherent scattering by $^1$H-rich samples, backgrounds levels were adjusted by a small constant such that the high-$q$ scattering displayed $q^{-4}$ dependence. The 0% D$_2$O sample was measured and analyzed in serial dilution on the ANSTO Bruker NanoStar SAXS instrument.

**Construction of A20$^{D315R}$, A20$^{R531D}$, A20$^{R531D,D315R}$, and A20$^{\Delta slo}$.** The A20 mutant strains harboring NADase D315R, SLO R531D, or *slo* deletion mutants were constructed by the allelic exchange mutagenesis[66]. The genes encoding NADase$_{D315R}$ and SLO$_{R531D}$ were cloned into the temperature-sensitive vector pCN143 that conferred spectinomycin resistance[67]. The resulting plasmids were utilized to construct A20$^{D315R}$ and A20$^{R531D}$. The pCN143 variant, which was inserted with a DNA fragment containing a chloramphenicol cassette flanked by the 506-bp upstream and 423-bp downstream sequences of *slo*, was built for the construction of A20$^{\Delta slo}$. The plasmids were respectively transferred into A20 by electroporation and the transformants were selected using spectinomycin at 30 °C. The transformants were transferred to 37 °C to select spectinomycin-resistant strains in which the plasmid was integrated into the chromosome through homologous recombination. Next, the resulting transformants in which plasmid excised from the chromosome via recombination were selected in antibiotic-free plates at 30 °C. The resulting strains harboring NADase D315R, SLO R531D, or *slo* deletion mutants were verified with DNA sequencing. For constructing A20$^{R531D,D315R}$, the SLO R531D mutation was introduced into A20$^{D315R}$.

**Measurement of NADase activity in GAS supernatants.** GAS cultures were collected when cells reached to an O.D.$_{600}$ of 0.6 in TSBY. The secreted NADase activity in culture supernatants of A20 and A20$^{D315R}$ and the measurement of NADase activity was performed as described above, except that activity of each sample was shown as the percentage of NAD$^+$ consumption relative to the TSBY medium-only control.

**Determination of secreted levels of NADase and SLO in the GAS cultures by immunoblot.** GAS cultures were collected when cells reached to an O.D.$_{600}$ of 0.6 in TSBY. Supernatants were collected, subjected to SDS-PAGE, and transferred to a polyvinylidene fluoride membrane. The membrane was blocked with 3% BSA for 1 h and probed with antibodies against NADase (1:1000, GTX64140, GeneTex) and SLO (1:1000, ab188539, Abcam). The blots were visualized by Western Lightning Plus-ECL (PerkinElmer).

**Detection of intracellular NADase by immunoblot.** The SLO-mediated translocation of NADase was evaluated[26]. Briefly, the A549 human lung epithelial cell line was infected with GAS at an MOI of 5 after 30 min of cytochalasin D treatment (0.6 µM) to block bacterial internalization. At 3.5 h-post-infection, the infected cells were treated with gentamicin (125 µg/ml) for 30 min and washed three times with sterilized PBS. The cells were subsequently resuspended in a lysis buffer consisting of 0.005% saponin in PBS and incubated for 20 min at 37 °C followed by centrifugation. The resulting supernatant was the cytoplasmic fraction, while the resulting pellet was the cell membrane fraction. The samples were analyzed by immunoblotting with anti-NADase (1:1000, GTX64140, GeneTex) and anti-SLO (1:1000, ab188539, Abcam). Antibodies against β-actin (1:10000, MAB1501, Sigma–Aldrich) and CD44 (1:2000, Cat. #3570; Cell Signaling Technology) were used as loading controls for the cytosolic and membrane fractions, respectively.

**Measurement of intracellular NADase activity.** The activity assay of NADase translocated into host cells after the GAS-A549 interaction was performed[16]. Briefly, the cytoplasmic fractions of GAS-infected A549 cells were two-fold serial diluted in 100 µl PBS buffer. Each sample was mixed with 200 µl of 1 mM β-NAD$^+$ solution (Sigma–Aldrich) and incubated at 37 °C for 1 h. Then, the reactions were terminated by the addition of 200 µl of 5 M NaOH. Following an additional 1-hour incubation at room temperature in the dark, the level of uncleaved β-NAD$^+$ in each reaction was determined by ELISA reader (360 nm absorbance; Multiskan FC,

Thermo). The NADase activity of each sample was reported as a relative percentage of $NAD^+$ consumption compared with the PBS buffer-only control.

**Cytotoxicity assay**. The cytotoxic effects of GAS strains on A549 cells were determined by measuring the levels of lactate dehydrogenase (LDH) released from GAS-infected cells. A549 cells were infected by GAS at different MOIs (5, 10, or 20) and then incubated for 4 h. The supernatants of the infected cell cultures were collected and subjected to the assay. The level of LDH was measured using LDH-Cytotoxicity Assay Kit-II (K313-500, BioVision), according to the manufacturer's instruction. The LDH levels of uninfected cells lysed with 1% Triton X-100 were served as 100% cell death with maximal LDH release.

**Intracellular survival assay**. The survival of GAS within U937 cells, a human macrophage cell line, was assessed[68]. Briefly, U937 cells were incubated with 15 nM of phorbol 12-myristate 13-acetate (PMA) for 3 days to induce differentiation from monocyte-like to macrophage-like cells[69]. Then, the differentiated U937 cells were infected with GAS at an MOI of 50 in FBS-free RPMI medium. At 30-minute post-infection, the infected U937 cells were treated with 100 μg/ml of gentamicin to kill the bacteria not phagocytosed by U937. The gentamicin in the culture was removed with a PBS wash after 30 min of incubation. At 2- and 4-hour post-infection, the cells were washed with PBS twice and lysed with 0.1% Triton X-100 for 5 min. The cell lysates were serial diluted and plated on TSBY plates to determine the counts of GAS that survived within the macrophages.

**Mouse infection model**. All mouse experiments were conducted under a protocol approved by Institutional Animal Care and Use Committee (IACUC), National Cheng Kung University (IACUC Approval No: 110309). The air pouch subcutaneous infection model was performed[46–48]. Briefly, 8–10 week-old female C57BL/6 mice ($n = 8$) were subcutaneously injected with 0.5 ml of sterile air to form an air pouch and were inoculated with $1 \times 10^8$ CFU of A20 or A20$^{D315R}$. The lesion area caused by the infection was photographed at 0-hour and 24-hour post-infection, and the lesion sizes were determined by analyzing the images with the ImageJ software[70]. To determine the bacterial burden in the animals after 24 h of infection, 1 ml of sterile PBS was injected into the air pouch of each mouse. The bacterium-containing PBS was aspirated from the pouches, serial diluted in fresh PBS, and plated on TSBY plates to determine the number of recovered bacteria. To determine the levels of IL-1β production in the lesion area, the skin lesions were excised from the mice for immunoblot analysis. Briefly, the frozen skin samples were homogenized and lysed with RIPA buffer plus protease inhibitors for 20 min on ice[71], and the lysates were centrifuged at 13,000 g for 30 min at 4 °C. The resulting supernatants were collected, mixed with SDS sample buffer, and analyzed by immunoblotting with antibodies against IL-1β (1:500, AF-401, R&D systems) and β-actin (1:5000, A5441, Sigma–Aldrich).

**Statistics and reproducibility**. Statistical analyses were performed using Graph-Pad Prism version 8 (GraphPad Software). Comparison between experimental groups was performed using the two-tailed unpaired $t$-test or two-tailed Mann-Whitney $U$ test as indicated. The results of in vitro assays are presented as mean ± standard deviation, and the data from the animal model are shown as median with interquartile range. $P$-value < 0.05 was considered as statistically significant. The number of repeats for each experiment is described in the associated figure legends.

**Reporting summary**. Further information on research design is available in the Nature Portfolio Reporting Summary linked to this article.

## Data availability

Atomic coordinate and structure factor for NADase$_{193-451,G330D}$/SLO$_{106-574}$ complex were deposited in the Protein Data Bank (https://www.rcsb.org/) with code 7WVH. Scattering profiles and models of NADase and NADase/SLO complex have been deposited in the Small Angle Scattering Biological Data Bank (https://www.sasbdb.org/) with codes SASDM47 and SASDM57. Source data for figures can be found in Supplementary Data. Uncropped and unedited blot images are shown in Supplementary Fig. 12.

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

## Acknowledgements

We thank the technical services provided by the "Synchrotron Radiation Protein Crystallography Facility of the National Core Facility Program for Biotechnology, Ministry of Science and Technology" and the "National Synchrotron Radiation Research Center", a national user facility supported by the Ministry of Science and Technology, Taiwan. SAXS data were collected at the SIBYLS beamline at the Advanced Light Source (Berkeley, CA) which is supported by the National Institutes of Health (P30 GM124169), and the Integrated Diffraction Analysis Technologies program of the US Department of Energy Office of Biological and Environmental Research. The Advanced Light Source is a national user facility operated by Lawrence Berkeley National Laboratory on behalf of the US Department of Energy under contract DE-AC02-05CH11231, Office of Basic Energy Sciences. We also acknowledge ANSTO for access to the neutron scattering instrumentation via proposal DB9048 and the provision of deuterated material under proposals 9053 and 9058. The National Deuteration Facility is partly supported by the National Collaborative Research Infrastructure Strategy—an initiative of the Australian Government. This research was supported by grants (MOST 109-2320-B-006 -066 and MOST 110-2320-B-006 -023) to S.W. and fellowships from Ministry of Science and Technology and Xin Miao Education Foundation of Taiwan to W.J.T. We also thank Drs. Miki Senda, Meng-Chiao Ho, Ching-Hao Teng, Nadendla Eswarkumar, and Yu-Yuan Hsiao for their expertise and assistance throughout our study.

## Author contributions

W.T. and S.W. designed research; W.T., Y.L., Y.S., M.H., A.D., A.W., K.W., C.W., R.K., C.K., C.H., and J.W. performed experiments; W.T., M.H., A.D., A.W., U.J., P.T., C.C., J.W., Y.L., C.L., T.S., and S.W. analyzed data; W.T., M.H., A.D., A.W., Y.L., T.S., and S.W. wrote the paper.

## Competing interests

The authors declare no competing interests.
