## [Peer Review File · Communications Biology]

Reviewers' comments:

Reviewer #1 (Remarks to the Author):

In "Structural basis underlying the synergism of NADase and SLO in the pathogenesis of Group A Streptococcus infection", Tsai et al examine the interaction of two GAS toxins. Their structural studies appear reasonable and justified. Based on these observations, they generate a point mutant in order to disrupt association between these toxins and to discover the importance of this in pathogenesis. It is the activity of this mutant on which a large portion of their argument lies, and which could use additional validation. The writing is overall quite strong and clear, but in specific areas is lacking in method and statistical detail. It would further benefit from additional analysis and discussion for a larger audience outside specialists interested in these toxins. For a broader point, this work in part shows these toxins binding is not required for their activity, nor virulence. While this is not a negative finding per se, it is deserving of greater discussion as it somewhat goes against the argument for the study of these proteins. Please see below for more detailed specific comments

The animal model methods are unclear, and potentially problematic. It was performed "as previously described" with a reference to Liu et al Front Micro, which does not describe similar methods. It also appears to be an underpowered study. Independent of concerns on how this experiment was done, it is not clear that it has any biological significance. There is no difference in bacterial replication, and the difference in lesion area is slight, and subject to interpretation. It appears they believe 'necrotic skin lesion' (which may just be the red scabbed area) is a specific indicator of disease, but no indication for this is given. Certainly, non-scabbed tissue seems to play a role in disease, so saying that specifically is important does not advance a useful argument. The lack of effect on bacterial replication or mouse survival does not support "enhanced pathogenicity of GAS is demonstrated in vitro and in vivo" (line 335). This rather argues the counter, that this mechanism has little to no effect, and is certainly neither essential nor sufficient.

Figure 4 is lacking in controls. First, negative controls of SLO-mutant and NADase mutant are essential to support the specificity of this system. Second, it is not clear that the small number of bacteria in 4C is because they are intracellular, as they could be persisters, adherent on the sides of wells, etc, as their specific location is not shown, just a failure to kill. Third, Translocation should be shown with their proteins, not just bacteria that may have additional differences. Fourth, this should include the NADase(D315R)/SLO(R531D) complex to show functional restoration to support their observations in other experiments (as in 2B). Last, the number of samples (unclear whether biological or technical replicates, and how many experiments) is variable for no clear reason.

The argument of line 245 "discrepant effect of cytochalasin D on bacterial invasion..." does not appear to make sense. In the protocol of 4D, rapid gent treatment means few living bacteria are present most of the time, where 4a has late gent treatment, so most of the activity is present throughout the experiment, but the kill at the end means a focus on a tiny subpopulation that likely has a negligible contribution to total activity. The framing of these experiments makes it unclear what specific argument is being advanced, but with some many different variables, and lots of experimental biases being introduced, its not clear that the mechanisms the authors are trying to connect are causal to their observations.

Methods are not described for much of the supplemental materials. S4 for example, appears to be describing activity of purified protein. If so, it is not clear that there is differences in expression or activity when expressed by GAS. S9 appears to be a Western blot, but the conditions are unclear. I'm assuming it could be GAS cultures treated as during an infection. The text claims these are comparable bands, but they look different, theres no loading or positive or negative controls, repeats, or quantification. If all this is controlled for, an effect on protein secretion can then be excluded, but this would be better examined during infection, where effects on protein stability and cell targeting can be examined. Binding of SLO and SPN to the host cell membrane (as in ref 25 and other papers) is ultimately the measure required to be sure differences from the mutation are solely interaction between these proteins and not other, undercharacterized, activities.

NADase activity is a virulence factor and is cytotoxic – if the difference in activity in Fig 4B is

biologically significant, why is there no effect on growth in vitro? Cell viability is an essential control for the experiments of Fig 4, as SLO and NADase are both cytotoxic.

Regarding Trp81 (discussed on line 22): Their model appears at odds with known biochemical observations. The speculation provided that dynamics may reveal this residue is not apparent in their structure. Further characterization of this region to support their model is needed.

Presumably, the NADase mutation does not impact IFS inhibition. Have the authors excluded effects from this interaction on bacterial viability and bacterial NAD levels?

Throughout, the numbers and statistics are completely missing or inadequate. Most measures in the reporting summary checklist are absent, including sample size, statistical tests, biological/technical replicates, etc, are absent.

Reviewer #2 (Remarks to the Author):

In the manuscript Tsia et. al. solves the structure of a truncated NADase/SLO complex and uncover how they interact with each other. In particular a salt bridge between NADase D315 and SLO R531 is important for the interaction. Further analysis reveals that the interaction and conformational dynamics between NADase and SLO is important for the function of the complex. In addition the authors demonstrate that disrupting the interaction between NADase and SLO affects GAS infection in cells and a mouse model. The manuscript presents novel, important results and is of interest to people involved in the field, but certain issues need to be resolved before it can be considered for publication.

The writing in the manuscript is not clear and it is difficult to follow what the authors are trying to communicate. The manuscript would greatly benefit if a native speaker worked on the writing and sentence formulation.

The X-ray data is fine, and the structure refinement has been performed in a satisfactory manner. The Rmerge value for the high resolution shell is a bit high at 0.54, ideally it should be under 0.5. The data analysis agrees with the solved structure.

The SAXS data is of high quality and analyzed correctly. The modeling based on the data also fits the data well.

The SANS data is also of high quality, but I am uncertain what they are trying to demonstrate with it. A better explanation of why the experiment was performed and what it shows would improve the manuscript.

In Supp fig 7. Legend states "The modelling indicates that 66 % is extended and 34 % compact" I do not know what the authors mean by modelling here, maybe they mean analysis. The experiments on A549 and U937 were carried out with very different MOIs. For the A549 cells the MOI was 5 and for the U937 cells the MOI was 50. Why is there such a huge difference between the MOIs?

The cytotoxicity experiments on U937 cells should be repeated with uninfected cells to determine what the baseline cytotoxicity level is, since the lactate dehydrogenase release will happen with any membrane damage.

For the mouse experiments the age is reported to be 8-10 weeks while in the reporting summary states that the age is 12 weeks.

For figure 7B the measurement of the lesion area is missing.

For supp fig 9 where the authors look at the secreted level of NADase and SLO between WT and mutated GAS should be repeated with a loading control.

Reviewer #3 (Remarks to the Author):

This manuscript described the critical role of D315 of NADase in forming the complex with SLO through X-ray crystal structure by forming the salt bridge with R531 of SLO. The mutation of D315 to Arg disrupted the complex formation. The in vitro and in vivo studies also supported the role of the complex formation of NADase/SLO complex in the pathogenicity of the group A Streptococcus (GAS). The X-ray crystal structure was only from the truncated NADase and SLO with 2.45 angstrom. The authors did study the interaction between full length NADase and SLO using SAXS and SANS. The authors demonstrated the formation of the complex between NADaseD315R/SLO531D, it would be interesting to see if a double mutant of GAS genome to NADaseD315R/SLO531D can restore the pathogenicity of GAS (or at least make some discussions). It is not clear why the authors chose G330D mutant for crystallization. It would be interesting to see the crystal structure of both wildtype and G330D mutant complex rather than through superimposing in Fig. S2. Some other minor issues: (1) it would be clearer if the authors define the symbol of Δ as the truncation or use NADase193-451 (same for SLO) in the manuscript; (2) give the full name of MALS (multi-angle light scattering?).

Point-by-point responses to the reviewer

Reviewers' comments:

Reviewer #1 (Remarks to the Author):

In “Structural basis underlying the synergism of NADase and SLO in the pathogenesis of Group A Streptococcus infection”, Tsai et al examine the interaction of two GAS toxins. Their structural studies appear reasonable and justified. Based on these observations, they generate a point mutant in order to disrupt association between these toxins and to discover the importance of this in pathogenesis. It is the activity of this mutant on which a large portion of their argument lies, and which could use additional validation. The writing is overall quite strong and clear, but in specific areas is lacking in method and statistical detail. It would further benefit from additional analysis and discussion for a larger audience outside specialists intreated in these toxins. For a broader point, this work in part shows these toxins binding is not required for their activity, nor virulence. While this is not a negative finding per se, it is deserving of greater discussion as it somewhat goes against the argument for the study of these proteins. Please see below for more detailed specific comments

The animal model methods are unclear, and potentially problematic. It was performed “as previously described” with a reference to Liu et al Front Micro, which does not describe similar methods. It also appears to be an underpowered study. Independent of concerns on how this experiment was done, it is not clear that it has any biological significance. There is no difference in bacterial replication, and the difference in lesion area is slight, and subject to interpretation. It appears they believe ‘necrotic skin lesion’ (which may just be the red scabbed area) is a specific indicator of disease, but no indication for this is given. Certainly, non-scabbed tissue seems to play a role in disease, so saying that specifically is important does not advance a useful argument. The lack of effect on bacterial replication or mouse survival does not support “enhanced pathogenicity of GAS is demonstrated in vitro and in vivo” (line 335). This rather argues the counter, that this mechanism has little to no effect, and is certainly neither essential nor sufficient.

Figure 4 is lacking in controls. First, negative controls of SLO-mutant and NADase mutant are essential to support the specificity of this system. Second, it is not clear that the small number of bacteria in 4C is because they are intracellular, as they could be persisters, adherent on the sides of wells, etc, as their specific location is not shown, just a failure to kill. Third, Translocation should be shown with their proteins, not just bacteria that may have additional differences. Fourth, this should include the NADase(D315R)/SLO(R531D) complex to show functional restoration to support their observations in other experiments (as in 2B). Last, the number of samples (unclear whether biological or technical replicates, and how many experiments) is variable for no clear reason.

The argument of line 245 “discrepant effect of cytochalasin D on bacterial invasion...” does not appear to make sense. In the protocol of 4D, rapid gent treatment means few living bacteria are present most of the time, where 4a has late gent treatment, so most of the activity is present

throughout the experiment, but the kill at the end means a focus on a tiny subpopulation that likely has a negligible contribution to total activity. The framing of these experiments makes it unclear what specific argument is being advanced, but with some many different variables, and lots of experimental biases being introduced, its not clear that the mechanisms the authors are trying to connect are causal to their observations.

Methods are not described for much of the supplemental materials. S4 for example, appears to be describing activity of purified protein. If so, it is not clear that there is differences in expression or activity when expressed by GAS. S9 appears to be a Western blot, but the conditions are unclear. I'm assuming it could be GAS cultures treated as during an infection. The text claims these are comparable bands, but they look different, theres no loading or positive or negative controls, repeats, or quantification. If all this is controlled for, an effect on protein secretion can then be excluded, but this would be better examined during infection, where effects on protein stability and cell targeting can be examined. Binding of SLO and SPN to the host cell membrane (as in ref 25 and other papers) is ultimately the measure required to be sure differences from the mutation are solely interaction between these proteins and not other, undercharacterized, activities.

NADase activity is a virulence factor and is cytotoxic – if the difference in activity in Fig 4B is biologically significant, why is there no effect on growth in vitro? Cell viability is an essential control for the experiments of Fig 4, as SLO and NADase are both cytotoxic.

Regarding Trp81 (discussed on line 22): Their model appears at odds with known biochemical observations. The speculation provided that dynamics may reveal this residue is not apparent in their structure. Further characterization of this region to support their model is needed.

Presumably, the NADase mutation does not impact IFS inhibition. Have the authors excluded effects from this interaction on bacterial viability and bacterial NAD levels?

Throughout, the numbers and statistics are completely missing or inadequate. Most measures in the reporting summary checklist are absent, including sample size, statistical tests, biological/technical replicates, etc, are absent.

Comment 1: The animal model methods are unclear, and potentially problematic. It was performed “as previously described” with a reference to Liu et al Front Micro, which does not describe similar methods.

Response: We apologize to the reviewer that the previously cited reference regarding the air-pouch animal model is overly brief that causes the confusion. In the revised manuscript, we have included two more references (Kuo *et al.* 1998; Lu *et al.* 2013) and added a more detailed description of the procedures in the Materials and Methods section (pages 24-25, lines 539-556).

Added references:

- 47 Kuo, C. F. *et al.* Role of streptococcal pyrogenic exotoxin B in the mouse model of group A streptococcal infection. *Infect Immun* **66**, 3931-3935, doi:10.1128/IAI.66.8.3931-3935.1998 (1998).
- 48 Lu, S. L. *et al.* Kallistatin modulates immune cells and confers anti-inflammatory response to protect mice from group A streptococcal infection. *Antimicrob Agents Chemother* **57**, 5366-5372, doi:10.1128/AAC.00322-13 (2013).

Comment 2: It also appears to be an underpowered study. Independent of concerns on how this experiment was done, it is not clear that it has any biological significance. There is no difference in bacterial replication, and the difference in lesion area is slight, and subject to interpretation. It appears they believe ‘necrotic skin lesion’ (which may just be the red scabbed area) is a specific indicator of disease, but no indication for this is given. Certainly, non-scabbed tissue seems to play a role in disease, so saying that specifically is important does not advance a useful argument. The lack of effect on bacterial replication or mouse survival does not support “enhanced pathogenicity of GAS is demonstrated in vitro and in vivo” (line 335). This rather argues the counter, that this mechanism has little to no effect, and is certainly neither essential nor sufficient.

Response: We agree with the reviewer that the animal experiments were underpowered. Thus, we repeated the air pouch animal experiments with increased animal numbers (N=8) to confirm the biological significance of the NADase/SLO complex in GAS virulence. At 24-hour post-infection, the A20^{D315R} mutant induced significantly smaller sizes of lesions than A20 (Fig. 5a and 5b, Supplementary Fig. 11). Consistent with the smaller lesion caused by A20^{D315R}, lower levels of bacterial counts were recovered from A20^{D315R} than those from A20 (Fig. 5c). Moreover, A20^{D315R}-infected animals produced lower levels of the inflammatory cytokine IL-1 β compared to A20-infected mice (Fig. 5d), suggesting that A20^{D315R}-infected animals exhibited lower levels of inflammation than the A20-infected ones. Taken together, these findings support that the formation of the NADase/SLO complex contributes to the pathogenesis of GAS. The updated results have been incorporated into the revised manuscript (page 15, lines 316-328).

Previously, we showed no significant difference in the bacterial burdens recovered from A20^{D315R}- and A20-infected mice (Fig. 5C in the previous version), while in the revised version we showed that the bacterial burden in A20^{D315R}-infected mice was significantly lower than those in A20-infected mice at 24-hour post-infection (Fig. 5c). We apologize that the bacterial counts shown in the previous version was actually recovered from 48-hour post-infection that was mislabeled as 24-hour post-infection. In the previous animal experiments, the images of skin lesions were photographed at 24-hour and 48-hour post-infection (please see the Figure below) and the bacterial burdens were determined at 48-hour post-infection before animal sacrificing. The infected mice may reach the recovered stage at 48-hour post-infection (Figure below), and thus the bacterial burdens and skin lesions in A20^{D315R}- and A20-infected mice showed no significant difference. Therefore, 24-hour post-infection is a better time window to compare the virulence of A20^{D315R} and A20.

Comment 3: First, negative controls of SLO-mutant and NADase mutant are essential to support the specificity of this system.

Response: We agree. In the revised manuscript, we have included the SLO-mutant of A20 (A20^{Δslo}), NADase mutant (A20^{D315R}), A20 and mock in the cytotoxicity and intracellular survival assays. As expected, A20^{Δslo} showed no significant NADase translocation into A549 cells after the bacterium-host interaction, in comparison to the mock-treated group (Fig. 4b and 4c). Also, A20^{Δslo} showed significantly lower survival within the macrophage cell line U937 than A20 and the NADase mutant A20^{D315R} (Fig. 4g).

Comment 4: Second, it is not clear that the small number of bacteria in 4C is because they are intracellular, as they could be persisters, adherent on the sides of wells, etc, as their specific location is not shown, just a failure to kill.

Response: We agree with the reviewer that the small number of bacteria are not necessarily located within the host cells. However, the low bacterial counts were not likely to affect the results of the translocation of NADase because A20^{Δslo} also showed a similar level of residual live bacteria in the same experiment settings, while no significant NADase translocation was observed (Fig. 4b and 4c). This indicates that the residual live bacteria did not play a significant role in affecting the results of the NADase translocation. Given that the Fig. 4C in the previous manuscript causes confusion and is not the main theme of the manuscript, we have removed it from the Result section.

Comment 5: Third, Translocation should be shown with their proteins, not just bacteria that may have additional differences.

Response: We agree and we have investigated the protein levels of the intracellular NADase after A549 cells were incubated with A20, A20^{D315R}, and A20^{Δslo} by immunoblots. Consistent with the results of the intracellular NADase activity, the A20-infected cells showed significantly higher levels of intracellular NADase than A20^{D315R}, while A20^{Δslo}-infected cells showed no significant NADase translocation into the cells. The results are included in the revised manuscript (Fig. 4b, page 13, lines 271-274) and the experimental procedures are described in the Materials and Methods section (pages 22-23, lines 493-505).

Comment 6: Fourth, this should include the NADase(D315R)/SLO(R531D) complex to show functional restoration to support their observations in other experiments (as in 2B).

Response: Thank you for the suggestion. To perform the restoration experiments, we have constructed the SLO_{R531D} mutant (A20^{R531D}) and NADase_{D315R}/SLO_{R531D} double mutant (A20^{R531D,D315R}) strains. Unexpectedly, significantly lower levels of SLO protein secretion were observed in A20^{R531D} (Supplementary Fig. 9d and 9e). Given that the translocation of NADase is SLO-dependent, the discrepancy in the SLO secretion levels between A20 and A20^{R531D,D315R} did not allow us to properly evaluate whether rescuing the salt bridge could restore the ability of NADase translocation in A20^{R531D,D315R}. On the other hand, A20^{R531D} and A20^{R531D,D315R} showed comparable levels of SLO secretion (Supplementary Fig. 9d and 9e) that permits us to investigate the correlation of salt bridge formation with NADase translocation. The results show A20^{R531D,D315R} exhibited a higher level of NADase translocation than A20^{R531D} during GAS-A549 infection (Fig. 4e), demonstrating the regeneration of salt bridge at NADase/SLO complex interface (Fig. 2b) correlates with functional restoration. The new results are incorporated into the Results section (Page 14, lines 289-305).

Comment 7: Last, the number of samples (unclear whether biological or technical replicates, and how many experiments) is variable for no clear reason.

Response: In the revised manuscript, the results in Figure 4 have been repeated. The data are the representative of three biologically independent experiments performed in quadruplicate.

Comment 8: The argument of line 245 “discrepant effect of cytochalasin D on bacterial invasion...” does not appear to make sense. In the protocol of 4D, rapid gent treatment means few living bacteria are present most of the time, where 4a has late gent treatment, so most of the activity is present throughout the experiment, but the kill at the end means a focus on a tiny subpopulation that likely has a negligible contribution to total activity. The framing of these experiments makes it unclear what specific argument is being advanced, but with some many different variables, and lots of experimental biases being introduced, its not clear that the mechanisms the authors are trying to connect are causal to their observations.

Response: We apologize for the confusion. Please allow us to explain in details. The purpose of the experiments is to reveal the biological significance of NADase/SLO complex formation in GAS cytotoxicity (Fig. 4A in the previous manuscript) and in intracellular survival in immune cells (Fig. 4D in the previous manuscript). Therefore, the timings of gentamicin treatment in the two protocols have their respective purpose.

The protocol described in Fig. 4A is to measure the SLO-mediated NADase translocation into host cells and the consequent cytotoxic effects. The reason for the late gentamicin treatment (Fig. 4A in the previous manuscript) is to allow the adequate period of time for the live bacteria to produce and secrete sufficient amount of NADase and SLO. Since the live bacteria may interfere with the measurement of NADase translocation, the late treatment of gentamicin is to kill the live

GAS that stay outside the host cells, while cytochalasin D treatment was to block the bacterium internalization by the host cells. Although after the retreatment of both gentamicin and cytochalasin D, some live bacteria still remained (Fig. 4C in the previous manuscript). However, the amounts of the remaining live bacteria were neglectable which did not interfere with the results and the Fig. 4C has been removed from the revised manuscript (response to comment 4). To avoid the confusion, the sentence “discrepant effect of cytochalasin D on bacterial invasion...” has also been removed in the revised version.

The protocol shown in the Fig. 4D of the previous version (Fig. 4f in the revised manuscript) is to measure the intracellular survival of GAS after the bacteria are phagocytosed by macrophages. The early gentamicin treatment was to kill the bacteria not phagocytosed by the macrophages, which allows the measurement of live bacterial amounts within macrophages (Fig. 4g in the revised manuscript).

Comment 9: Methods are not described for much of the supplemental materials. S4 for example, appears to be describing activity of purified protein. If so, it is not clear that there is differences in expression or activity when expressed by GAS. S9 appears to be a Western blot, but the conditions are unclear. I’m assuming it could be GAS cultures treated as during an infection. The text claims these are comparable bands, but they look different, there’s no loading or positive or negative controls, repeats, or quantification. If all this is controlled for, an effect on protein secretion can then be excluded, but this would be better examined during infection, where effects on protein stability and cell targeting can be examined. Binding of SLO and SPN to the host cell membrane (as in ref 25 and other papers) is ultimately the measure required to be sure differences from the mutation are solely interaction between these proteins and not other, undercharacterized, activities.

Response: We apologize and we have added the missing materials and methods in the revised Supplementary Information, including recombinant NADase activity assay, activity of NADase in GAS culture supernatant and the immunoblotting of NADase secretion (pages 2-4, lines 24-77 in Supplementary Information). Regarding the NADase secretion, we have repeated the experiment (Supplementary Fig. 9a in the revised manuscript) to detect and quantitate the secretion level by immunoblots with a loading control (the total protein of the bacterial lysates in the corresponding culture). Quantification of three independent immunoblots shows A20 and A20^{D315R} express similar levels of secreted NADase in their culture supernatant (Supplementary Fig. 9b in the revised manuscript). In addition, we found similar levels of NADase in A20- and A20^{D315R}-infected cells were bound to host cell membranes (Fig. 4b in the revised manuscript), suggesting D315R mutation on NADase did not affect the membrane-binding ability of NADase. We also found a similar membrane-binding level of SLO in A20- and A20^{D315R}- infected cells (Fig. 4b in the revised manuscript). Thus, we believe that the difference in NADase translocation into host cytosol between A20 and A20^{D315R} is mainly due to interference of the NADase-SLO interaction caused by the D315R mutation.

Comment 10: NADase activity is a virulence factor and is cytotoxic – if the difference in activity in Fig 4B is biologically significant, why is there no effect on growth in vitro? Cell viability is an essential control for the experiments of Fig 4, as SLO and NADase are both cytotoxic.

Response: We agree with the reviewer that only minor difference was detected between the cytotoxicity induced by A20 and A20^{D315R} at MOI=5. However, when we increased the MOI to 20, the statistically significant difference between A20^{D315R}- and A20-induced cytotoxicity was observed (Fig. 4d in the revised manuscript). In addition, the *slo* mutant A20^{Δslo} showed significantly lower cytotoxicity than A20 and A20^{D315R}. These findings suggest that the decreased NADase translocation into the A20^{D315R}-infected cells contributed to the decreased cytotoxic effect of A20^{D315R}. The new results are incorporated into the Results section (pages 13-14, lines 280-289).

Comment 11: Regarding Trp81 (discussed on line 22): Their model appears at odds with known biochemical observations. The speculation provided that dynamics may reveal this residue is not apparent in their structure. Further characterization of this region to support their model is needed.

Response: Recently, the structure of NADase N-terminal domain determined by NMR shows the side-chain of Trp81 is not solvent-exposed and thus is unavailable for glycan binding (Velarde *et al.* 2022). The reported study also further characterized four putative glycan binding residues of NADase (Y120, Y121, Y151, and H177). With the availability of this newly determined structure, we re-performed SAXS-based modeling on full-length NADase and NADase/SLO complex (Fig. 3). We also show that the putative glycan binding residues of NADase are solvent-exposed in the structural ensemble of the full-length NADase/SLO complex (Supplementary Fig. 8b), indicating the competence of the complex in receptor binding. In the revised manuscript, SAXS results have been updated (pages 9-12, lines 179-250).

Reference:

Velarde, J. J. *et al.* Structure of the *Streptococcus pyogenes* NAD⁺ Glycohydrolase Translocation Domain and Its Essential Role in Toxin Binding to Oropharyngeal Keratinocytes. *J Bacteriol* **204**, e0036621, doi:10.1128/JB.00366-21 (2022).

Comment 12: Presumably, the NADase mutation does not impact IFS inhibition. Have the authors excluded effects from this interaction on bacterial viability and bacterial NAD levels?

Response: Based on the crystal structure of NADase C-terminal domain in complex with IFS (Smith *et al.* 2011), NADase D315 is not involved in NADase-IFS interaction (Figure below, Left) and thus the mutation on D315 should have no effects on IFS inhibition. To confirm this, we compared the growth curves of A20 and A20^{D315R} that show the similar viability (Figure below, right). This suggests the activity of NADase in A20^{D315R} is inhibited by IFS similarly to A20. Given that NAD level affects bacterial viability, the similar viabilities of A20 and A20^{D315R} suggest that the bacterial NAD levels in the two strains are similar.

Reference:

Smith, C. L. *et al.* Structural basis of *Streptococcus pyogenes* immunity to its NAD⁺ glycohydrolase toxin. *Structure* **19**, 192-202, doi:10.1016/j.str.2010.12.013 (2011).

Comment 13: Throughout, the numbers and statistics are completely missing or inadequate. Most measures in the reporting summary checklist are absent, including sample size, statistical tests, biological/technical replicates, etc, are absent.

Response: We apologize for this. All the required information has been included in the revised reporting summary.

Reviewer #2 (Remarks to the Author):

In the manuscript Tsia et. al. solves the structure of a truncated NADase/SLO complex and uncover how they interact with each other. In particular a salt bridge between NADase D315 and SLO R531 is important for the interaction. Further analysis reveals that the interaction and conformational dynamics between NADase and SLO is important for the function of the complex. In addition the authors demonstrate that disrupting the interaction between NADase and SLO affects GAS infection in cells and a mouse model. The manuscript presents novel, important results and is of interest to people involved in the field, but certain issues need to be resolved before it can be considered for publication.

The writing in the manuscript is not clear and it is difficult to follow what the authors are trying to communicate. The manuscript would greatly benefit if a native speaker worked on the writing and sentence formulation.

The X-ray data is fine, and the structure refinement has been performed in a satisfactory manner. The Rmerge value for the high resolution shell is a bit high at 0.54, ideally it should be under 0.5. The data analysis agrees with the solved structure.

The SAXS data is of high quality and analyzed correctly. The modeling based on the data also fits the data well.

The SANS data is also of high quality, but I am uncertain what they are trying to demonstrate with it. A better explanation of why the experiment was performed and what it shows would improve the manuscript.

In Supp fig 7. Legend states “The modelling indicates that 66 % is extended and 34 % compact” I do not know what the authors mean by modelling here, maybe they mean analysis. The experiments on A549 and U937 were carried out with very different MOIs. For the A549 cells the MOI was 5 and for the U937 cells the MOI was 50. Why is there such a huge difference between the MOIs?

The cytotoxicity experiments on U937 cells should be repeated with uninfected cells to determine what the baseline cytotoxicity level is, since the lactate dehydrogenase release will happen with any membrane damage.

For the mouse experiments the age is reported to be 8-10 weeks while in the reporting summary states that the age is 12 weeks.

For figure 7B the measurement of the lesion area is missing.

For supp fig 9 where the authors look at the secreted level of NADase and SLO between WT and mutated GAS should be repeated with a loading control.

Comment 1: The writing in the manuscript is not clear and it is difficult to follow what the authors

are trying to communicate. The manuscript would greatly benefit if a native speaker worked on the writing and sentence formulation.

Response: The resubmitted version of our manuscript has been extensively revised by co-authors who are native speakers of English. Changes made in the manuscript are marked in yellow highlights.

Comment 2: The SANS data is also of high quality, but I am uncertain what they are trying to demonstrate with it. A better explanation of why the experiment was performed and what it shows would improve the manuscript.

Response: We agree and we have attempted to better introduce our motivation for SANS studies in the revised manuscript (pages 10-11, lines 208-222): “In addition to SAXS studies, we attempted to visualize each protein within the NADase/SLO complex by SANS with contrast variation, where the measurements were designed such that SLO would be contrast matched in 42% D₂O and the deuterium labelled NADase would be contrast matched in 100% D₂O. The latter measurement was of particular importance as it would yield direct insight into the conformation of SLO while bound to NADase. Unfortunately, the 100% D₂O sample showed signs of aggregation/oligomerization, which made interpretation of this contrast point infeasible. Nonetheless, the other SANS contrast points measured from the deuterated-NADase/SLO complex were successfully analyzed in terms of the same multistate model used to fit the SAXS data, and it was found that the contrast variation data was in excellent agreement with the same relative proportions of extended and compact complex, plus free NADase (Supplementary Fig. 7 and Supplementary Table 2). Thus, the SANS provides additional support that the NADase/SLO complex exhibits dynamic behavior, involving a rigid NADase/SLO core structure with dynamic peripheral domains.”

Comment 3: In Supp fig 7. Legend states “The modelling indicates that 66 % is extended and 34 % compact” I do not know what the authors mean by modelling here, maybe they mean analysis.

Response: In response for comment 11 from Reviewer 1, the SAXS modelling was repeated. With this new model in hand, the analysis of the SANS data was repeated, where the relative proportions of 9% extended, 53% compact and 38% free NADase determined using SAXS were found to be consistent with the SANS data. The figure caption has been updated with these numbers and the mention of “modelling” has been removed.

Comment 4: The experiments on A549 and U937 were carried out with very different MOIs. For the A549 cells the MOI was 5 and for the U937 cells the MOI was 50. Why is there such a huge difference between the MOIs?

Response: The experiments using the A549 epithelial cells and the U937 macrophage cells measured distinct pathogenic traits of the bacteria. Therefore, different MOIs were utilized. The A549 cells were utilized to evaluate the cytotoxicity of GAS strains, while U937 was utilized to

evaluate the intracellular survival of GAS phagocytosed by immune cells. To measure the intracellular survival, adequate amounts of bacteria are needed to be internalized by U937, so that we can measure the potential difference in the intracellular survivability of the bacteria. Therefore, in the experiments using U937, we utilized high MOI to increase the bacterial amounts internalized by the cells.

Comment 5: The cytotoxicity experiments on U937 cells should be repeated with uninfected cells to determine what the baseline cytotoxicity level is, since the lactate dehydrogenase release will happen with any membrane damage.

Response: We agree and have included uninfected cells and a negative control (*A20^{Δslo}*) in the repeated cytotoxicity experiments on U937 cells (Supplementary Fig. 10 in the revised manuscript).

Comment 6: For the mouse experiments the age is reported to be 8-10 weeks while in the reporting summary states that the age is 12 weeks.

Response: This was our mistake. The age of mice has been corrected to be 8-10 weeks in the revised reporting summary.

Comment 7: For figure 7B the measurement of the lesion area is missing.

Response: We apologize and we have included the description in the revised Material and Methods section (page 25, lines 544-547): “The lesion area caused by the infection was photographed at 0-hour and 24-hour post-infection, and the lesion sizes were determined by analyzing the images with the ImageJ software⁶⁹.”

One reference is added:

69 Schindelin, J., Rueden, C. T., Hiner, M. C. & Eliceiri, K. W. The ImageJ ecosystem: An open platform for biomedical image analysis. *Mol Reprod Dev* **82**, 518-529, doi:10.1002/mrd.22489 (2015).

Comment 8: For supp fig 9 where the authors look at the secreted level of NADase and SLO between WT and mutated GAS should be repeated with a loading control.

Response: We have repeated the immunoblots using the total protein of the GAS lysates in the corresponding culture as a loading control (Supplementary Fig. 9a in the revised manuscript).

Reviewer #3 (Remarks to the Author):

This manuscript described the critical role of D315 of NADase in forming the complex with SLO through X-ray crystal structure by forming the salt bridge with R531 of SLO. The mutation of D315 to Arg disrupted the complex formation. The in vitro and in vivo studies also supported the role of the complex formation of NADase/SLO complex in the pathogenicity of the group A Streptococcus (GAS). The X-ray crystal structure was only from the truncated NADase and SLO with 2.45 angstrom. The authors did study the interaction between full length NADase and SLO using SAXS and SANS. The authors demonstrated the formation of the complex between NADaseD315R/SLOR531D, it would be interesting to see if a double mutant of GAS genome to NADaseD315R/SLOR531D can restore the pathogenicity of GAS (or at least make some discussions). It is not clear why the authors chose G330D mutant for crystallization. It would be interesting to see the crystal structure of both wildtype and G330D mutant complex rather than through superimposing in Fig. S2. Some other minor issues: (1) it would be clearer if the authors define the symbol of Δ as the truncation or use NADase193-451 (same for SLO) in the manuscript; (2) give the full name of MALS (multi-angle light scattering?).

Comment 1: This manuscript described the critical role of D315 of NADase in forming the complex with SLO through X-ray crystal structure by forming the salt bridge with R531 of SLO. The mutation of D315 to Arg disrupted the complex formation. The in vitro and in vivo studies also supported the role of the complex formation of NADase/SLO complex in the pathogenicity of the group A Streptococcus (GAS). The X-ray crystal structure was only from the truncated NADase and SLO with 2.45 angstrom. The authors did study the interaction between full length NADase and SLO using SAXS and SANS. The authors demonstrated the formation of the complex between NADaseD315R/SLOR531D, it would be interesting to see if a double mutant of GAS genome to NADaseD315R/SLOR531D can restore the pathogenicity of GAS (or at least make some discussions).

Response: In the revised manuscript, we have constructed the SLO_{R531D} mutant (A20^{R531D}) and SLO_{R531D}/NADase_{D315R} double mutant (A20^{R531D,D315R}) strains to perform the restoration experiments. The results demonstrate the regeneration of the salt bridge at the NADase/SLO complex interface can restore the function of NADase translocation (Fig. 4e). This comment is similar to comment 6 made by Reviewer 1 and please see our previous response for details.

Comment 2: It is not clear why the authors chose G330D mutant for crystallization. It would be interesting to see the crystal structure of both wildtype and G330D mutant complex rather than through superimposing in Fig. S2.

Response: Our attempts to crystallize wild-type NADase in complex with SLO were not successful. NADase G330D is a common polymorphism and has been reported to bind SLO similarly to the wild type (Velarde *et al.* 2017). We designed various constructs using both wild-type and G330D mutants for complex crystallization. Crystallization of a protein complex is often challenging. With extensive efforts, crystals only grew from the G330D mutant complex. We agree with the reviewer that the reason for choosing the G330D mutant is not clear and we have

attempted to clarify it by describing the G330D polymorphism in the Introduction section (page 5, lines 91-96): “A common polymorphism G330D, with the presence of aspartate instead of glycine at position 330 of NADase, lacks detectable NADase activity³⁴. However, NADase_{G330D} remains a potent virulence factor and is able to interact with SLO similarly to the wild type¹⁷, suggesting the molecular evolution of NADase has a tendency for preservation of the NADase/SLO complex.”

Reference:

Velarde, J. J., O’Seaghdha, M., Baddal, B., Bastiat-Sempe, B. & Wessels, M. R. Binding of NAD⁺-Glycohydrolase to Streptolysin O Stabilizes Both Toxins and Promotes Virulence of Group A *Streptococcus*. *mBio* **8**, e01382-17, doi:10.1128/mBio.01382-17 (2017).

Comment 3: Some other minor issues: (1) it would be clearer if the authors define the symbol of Δ as the truncation or use NADase₁₉₃₋₄₅₁ (same for SLO) in the manuscript; (2) give the full name of MALS (multi-angle light scattering?).

Response: We agree. We have used NADase₁₉₃₋₄₅₁ and SLO₁₀₆₋₅₇₄ throughout the entire manuscript and have spelled out MALS (page 10, line 206).

REVIEWERS' COMMENTS:

Reviewer #1 (Remarks to the Author):

Tsai et al. have addressed all my prior concerns.

Reviewer #2 (Remarks to the Author):

The authors have successfully addressed the issues and I now recommend that the manuscript should be published.

Point-by-point responses to the reviewer

Reviewers' comments:

Reviewer #1 (Remarks to the Author):

Tsai et al. have addressed all my prior concerns.

Reviewer #2 (Remarks to the Author):

The authors have successfully addressed the issues and I now recommend that the manuscript should be published.

Response: We thank the reviewers for their time in reviewing our manuscript.